# The derlin Dfm1 couples retrotranslocation of a folded protein domain to its proteasomal degradation

Daniela G. Vitali[1]* , Daniel Fonseca[1]* , and Pedro Carvalho[1]

Endoplasmic reticulum (ER) proteins are degraded by proteasomes in the cytosol through ER-associated degradation (ERAD). This process involves the retrotranslocation of substrates across the ER membrane, their ubiquitination, and membrane extraction by the Cdc48/Npl4/Ufd1 ATPase complex prior to delivery to proteasomes for degradation. How the presence of a folded luminal domain affects substrate retrotranslocation and this event is coordinated with subsequent ERAD steps remains unknown. Here, using a model substrate with a folded luminal domain, we showed that Cdc48 ATPase activity is sufficient to drive substrate retrotranslocation independently of ERAD membrane components. However, the complete degradation of the folded luminal domain required substrate-tight coupling of retrotranslocation and proteasomal degradation, which was ensured by the derlin Dfm1. Mutations in Dfm1 intramembrane rhomboid-like or cytosolic Cdc48-binding regions resulted in partial degradation of the substrate with accumulation of its folded domain. Our study revealed Dfm1 as a critical regulator of Cdc48-driven retrotranslocation and highlights the importance of coordinating substrate retrotranslocation and degradation during ERAD.

## Introduction

Proteins in the endoplasmic reticulum (ER) are degraded by ER-associated protein degradation (ERAD) (Christianson and Carvalho, 2022; Wu and Rapoport, 2018). This process targets a broad range of substrates, including misfolded membrane and secretory proteins generated as byproducts of biosynthetic processes, as well as a set of folded ER proteins, mostly rate-limiting enzymes of lipid metabolism as part of homeostatic regulatory mechanisms. Thus, ERAD controls both protein and lipid homeostasis in the ER and has a central role in cellular physiology in health and disease (Needham et al., 2019; Bhattacharya and Qi, 2019).

ERAD is carried out by multiple, membrane integral ubiquitin ligase complexes, each with specificity for distinct substrate classes (Christianson and Carvalho, 2022; Krshnan et al., 2022). In yeast, there are three ERAD ubiquitin ligase complexes—the Hrd1, Doa10, and Asi complexes—and despite their different substrate specificities, the general steps leading to substrate degradation are similar (Wu and Rapoport, 2018). Upon recognition and engagement with an ERAD ubiquitin ligase complex, substrates are transported across the ER membrane into the cytosol, a process called retrotranslocation. Once exposed on the cytosolic surface of the ER, substrates are polyubiquitinated and subsequently extracted from the ER membrane through the activity of the Cdc48 complex, which includes

the hexameric Cdc48 ATPase as well as Npl4 and Ufd1 cofactors (Bays et al., 2001; Ye et al., 2001; Jarosch et al., 2002; Stein et al., 2014; Bodnar and Rapoport, 2017; Twomey et al., 2019). Several adaptor proteins, including the multiubiquitination enzyme Ufd2 and the shuttle factors Rad23 and Dsk2, finally hand the ubiquitinated substrate to the proteasome for degradation (Medicherla et al., 2004; Richly et al., 2005; Baek et al., 2011).

Despite this general scheme, the molecular mechanisms of the individual ERAD steps are less clear, in particular substrate retrotranslocation across the ER membrane. In vivo and in vitro studies suggest that for ER luminal substrates, the major retrotranslocation component is the Hrd1 ubiquitin ligase (Carvalho et al., 2010; Baldridge and Rapoport, 2016; Vasic et al., 2020; Wu et al., 2020). Hrd1-mediated retrotranslocation depends on its membrane domain, which recent cryo-EM studies revealed to display features reminiscent of other protein-conducting channels such as the Sec translocon that transports unfolded, newly synthesized proteins into the ER lumen (Schoebel et al., 2017; Wu et al., 2020). The yeast derlins Der1 and Dfm1, members of the rhomboid protein superfamily (Freeman, 2014), have also been implicated in retrotranslocation (Mehnert et al., 2014; Neal et al., 2018; Wu et al., 2020; Nejatfard et al., 2021). Despite their similarity, Der1 and Dfm1 appeared to have distinct substrate specificity. While Der1 interacts with and is required for the

---

[1]Sir William Dunn School of Pathology, University of Oxford, Oxford, UK.

*D.G. Vitali and D. Fonseca contributed equally to this paper.   Correspondence to Pedro Carvalho: pedro.carvalho@path.ox.ac.uk.

degradation of misfolded ER luminal substrates (Knop et al., 1996; Carvalho et al., 2006; Denic et al., 2006), Dfm1 appears to target mostly misfolded membrane proteins (Stolz et al., 2010; Neal et al., 2018). The basis for these substrate preferences as well as the precise mechanisms by which ERAD components facilitate retrotranslocation remain obscure. The picture is even less clear for the retrotranslocation of ERAD substrates containing folded domains. It is well established that folded domains within the ER lumen slow down the degradation of ERAD substrates, suggesting that their retrotranslocation is challenging (Tirosh et al., 2003; Bhamidipati et al., 2005; Carvalho et al., 2010; Shi et al., 2019). However, how such domains are retrotranslocated and whether the process requires additional components is unknown.

Most ERAD substrates characterized so far are degraded constitutively and exist in cells as a heterogeneous ensemble of intermediates. Constitutive degradation of substrates also precludes the uncoupling between protein folding and retrotranslocation. Here, we circumvented these problems by employing an inducible auxin-based degron system. By uncoupling protein folding from subsequent steps, we searched for factors required for the retrotranslocation and degradation of a model substrate with a luminal folded domain. Our data identified the Cdc48 ATPase as an essential component at two distinct steps that involve different cofactors and ATPase activity requirements. In contrast, most known ERAD membrane factors were dispensable. The notable exception was the derlin Dfm1, which was critical in coupling the retrotranslocation of the folded domain with its degradation by the proteasome. In the absence of Dfm1, Cdc48-driven substrate retrotranslocation resulted in incomplete degradation and accumulation of its folded domain. These findings uncover a new function for Dfm1 in regulating Cdc48-mediated membrane substrate extraction and highlight the importance of coordinating substrate retrotranslocation and degradation during ERAD.

## Results

### An auxin-induced assay to study ER membrane protein degradation

Generation of dihydrofolate reductase (DHFR) fusion proteins is a well-established approach to study protein translocation across lipid bilayers (Eilers and Schatz, 1986; Arkowitz et al., 1993; Gambill et al., 1993; Tirosh et al., 2003; Bhamidipati et al., 2005; Shi et al., 2019). To study how the presence of folded luminal domains on ER membrane proteins impacts their retrotranslocation and subsequent proteasomal degradation, we generated a chimeric membrane protein in the ER lumen containing the *Escherichia coli* DHFR, which adopts a tightly folded conformation irrespective of the presence of folate ligands (Lee et al., 2001; Bhamidipati et al., 2005). The chimeric membrane protein was engineered to adopt a type I topology in the ER membrane with the N-terminal signal sequence of the ER chaperone Kar2 followed by a triple HA tag, an opsin glycosylation site, the DHFR domain, and the transmembrane segment of Ost1, an ER protein. The cytosolic domain of the chimera contained four tandem repeats of the 44-amino acid minimal degron (AID*)

derived from the *Arabidopsis thaliana* IAA17 protein (Morawska and Ulrich, 2013), followed by nine repeats of the MYC epitope. The chimeric construct was termed inducible Retro-Clogger (iRC) for the reasons explained below (Fig. 1 A).

Using fractionation experiments, we established that the iRC substrate was inserted in the ER membrane (Fig. 1 B). Moreover, an iRC derivative fused to the green fluorescent protein, GFP-iRC, showed a distribution consistent with ER localization (Fig. 1 C). To confirm that iRC acquired the correct topology, we examined whether the opsin sequence in iRC was glycosylated. Treatment of yeast lysates with peptide:N-glycosidase F (PNGaseF), which digests the N-glycans, resulted in faster iRC migration by SDS-PAGE (Fig. 1 D). Thus, iRC is targeted to the ER membrane with its N-terminus facing the ER lumen (Fig. 1 E).

ER ubiquitin ligases, such as Hrd1, are required for both ubiquitination and retrotranslocation of ERAD substrates (Carvalho et al., 2010; Baldridge and Rapoport, 2016; Vasic et al., 2020; Wu et al., 2020). To study substrate retrotranslocation bypassing the need for ubiquitination by the endogenous ER-resident ubiquitin ligases, we exploited the plant-based auxin-induced degradation (AID) (Nishimura et al., 2009; Morawska and Ulrich, 2013). The AID system relies on the expression of a plant-specific F-box protein, Tir1, to recognize AID* motifs in substrates only in the presence of auxin (indole-3-acetic acid, or IAA), triggering their polyubiquitination and degradation (Fig. 1 E).

In yeast cells expressing constitutively the *A. thaliana* Tir1 (*At*Tir1) and in the absence of auxin, iRC is a long-lived protein. Even upon inhibition of protein synthesis with cycloheximide (CHX), there is no appreciable iRC turnover over a period of 60 min (Fig. 1 F). Upon auxin addition, the iRC was rapidly degraded irrespective of the presence of cycloheximide (without cycloheximide $t_{1/2}$ = 15 min; with cycloheximide $t_{1/2}$ = 10 min). Importantly, iRC degradation was selective even at higher auxin concentrations (Fig. S1 A) and dependent on the activity of the proteasome (Fig. 1 G). Thus, auxin-induced ubiquitination of iRC is sufficient to trigger its retrotranslocation and proteasomal degradation in an acute and synchronous manner.

Long exposures of the blots revealed that while most iRC was degraded upon auxin addition, small amounts of an ~40 kD fragment appeared concomitantly with the decrease in full-length iRC. Accumulation of this ~40 kD fragment became even more prominent in cells expressing iRC from the strong *ADH1* promoter (*ADHp*) instead of the weak *PRC1* promoter (*CPYp*) (Fig. 1 G). Based on the size and reactivity with the HA antibody, this fragment likely corresponds to tightly folded DHFR, suggesting that the degradation of this domain is rate-limiting. Moreover, the disappearance of the MYC tag indicates that iRC degradation occurs from the C-terminus in the cytosol toward the N-terminus, initially in the ER lumen (Fig. S1 B).

### iRC degradation does not require ERAD ubiquitin ligases

Increasing evidence implicates ERAD ubiquitin ligases in substrate retrotranslocation (Carvalho et al., 2010; Baldridge and Rapoport, 2016; Wu et al., 2020; Vasic et al., 2020; Schmidt et al., 2020; Natarajan et al., 2020). Therefore, we tested whether the ubiquitin ligases Hrd1, Doa10, and Asi1 were

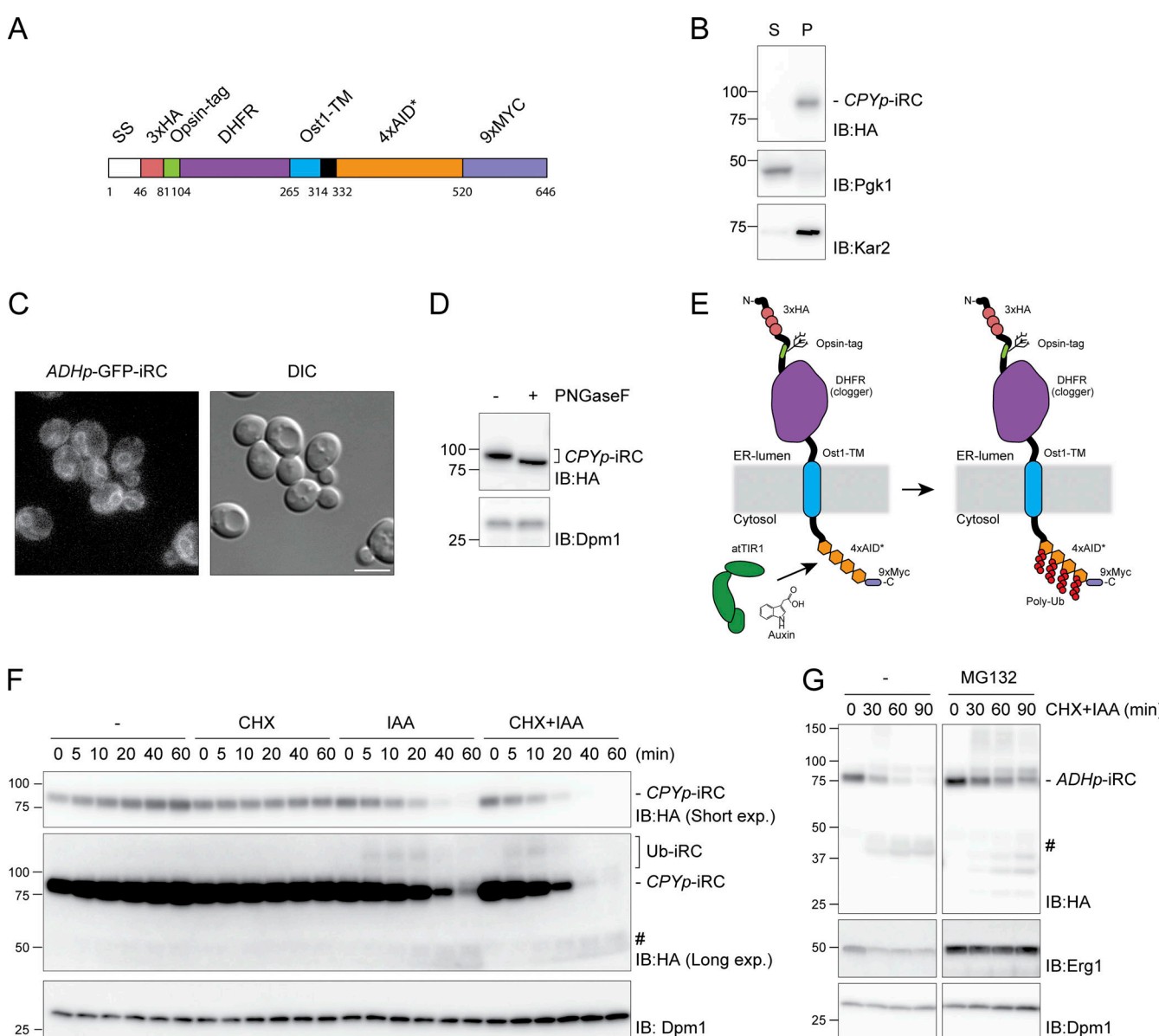

Figure 1. **Inducible Retro-Clogger (iRC), an inducible substrate to study protein retrotranslocation. (A)** Scheme of the inducible Retro-Clogger (iRC) substrate. The individual modules that compose the iRC are indicated. Numbers indicate the amino acid borders of each module. **(B)** iRC is membrane-associated. Crude membrane (M) and soluble (S) fractions from WT cells expressing iRC were prepared and analyzed by SDS-PAGE and Western blotting using an α-HA antibody. The ER protein Kar2 and the cytosolic phosphoglycerate kinase (Pgk1) proteins were used as controls and detected with α-Kar2 and Pgk1 antibodies, respectively. **(C)** Florescent microscopy images of WT cells expressing a GFP-tagged iRC version and corresponding DIC. Scale bar: 5 µm. **(D)** The iRC adopts a type I topology with the N-terminus in the ER lumen. Whole-cell extracts of WT cells expressing iRC were treated with PNGase F and analyzed by SDS-PAGE and Western blotting, respectively, using an α-HA antibody. The non-glycosylated dolichol phosphate mannose synthase (Dpm1) was used as a loading control and detected with the α-Dpm1 antibody. **(E)** Schematic representation of the topology of the iRC chimera and of its auxin-dependent ubiquitination. **(F)** Time-course analysis of iRC stability in cells expressing *At*Tir1 under the indicated conditions. Auxin (IAA) and cycloheximide (CHX) were added at time 0 and were used at 0.4 mM and 125 µg/ml, respectively. Note that the addition of IAA leads to the appearance of a faint high molecular weight smear consistent with iRC ubiquitination (Ub-iRC in long exposure). There is also the appearance of a faint low molecular fragment in late timepoints after IAA addition (#). Whole-cell extracts were analyzed by SDS-PAGE and Western blotting. iRC was detected with α-HA antibody. Dpm1 was used as a loading control and detected with the α-Dpm1 antibody. **(G)** iRC is degraded by the proteasome. *pdr5Δ* cells were treated with the proteasome inhibitor MG132 (25 µg/ml) for 30 min, and subsequently with IAA and CHX to follow iRC degradation. Samples were analyzed as in F. Source data are available for this figure: SourceData F1.

necessary for iRC retrotranslocation and degradation. Individual deletion of Asi1, Hrd1, and Doa10 did not prevent auxin-induced degradation of iRC (Fig. 2 A). Moreover, the degradation of iRC was unaffected by the simultaneous deletion of the three ubiquitin ligases or the ubiquitin-conjugating enzyme Ubc7 involved

in all ERAD branches (Fig. 2 A). As in wild-type (WT) cells, induction of iRC degradation in ERAD ubiquitin ligase mutants resulted in minute accumulation of the ~40 kD fragment. Therefore, auxin-induced degradation of the iRC substrate is independent of the canonical ERAD ubiquitin ligases.

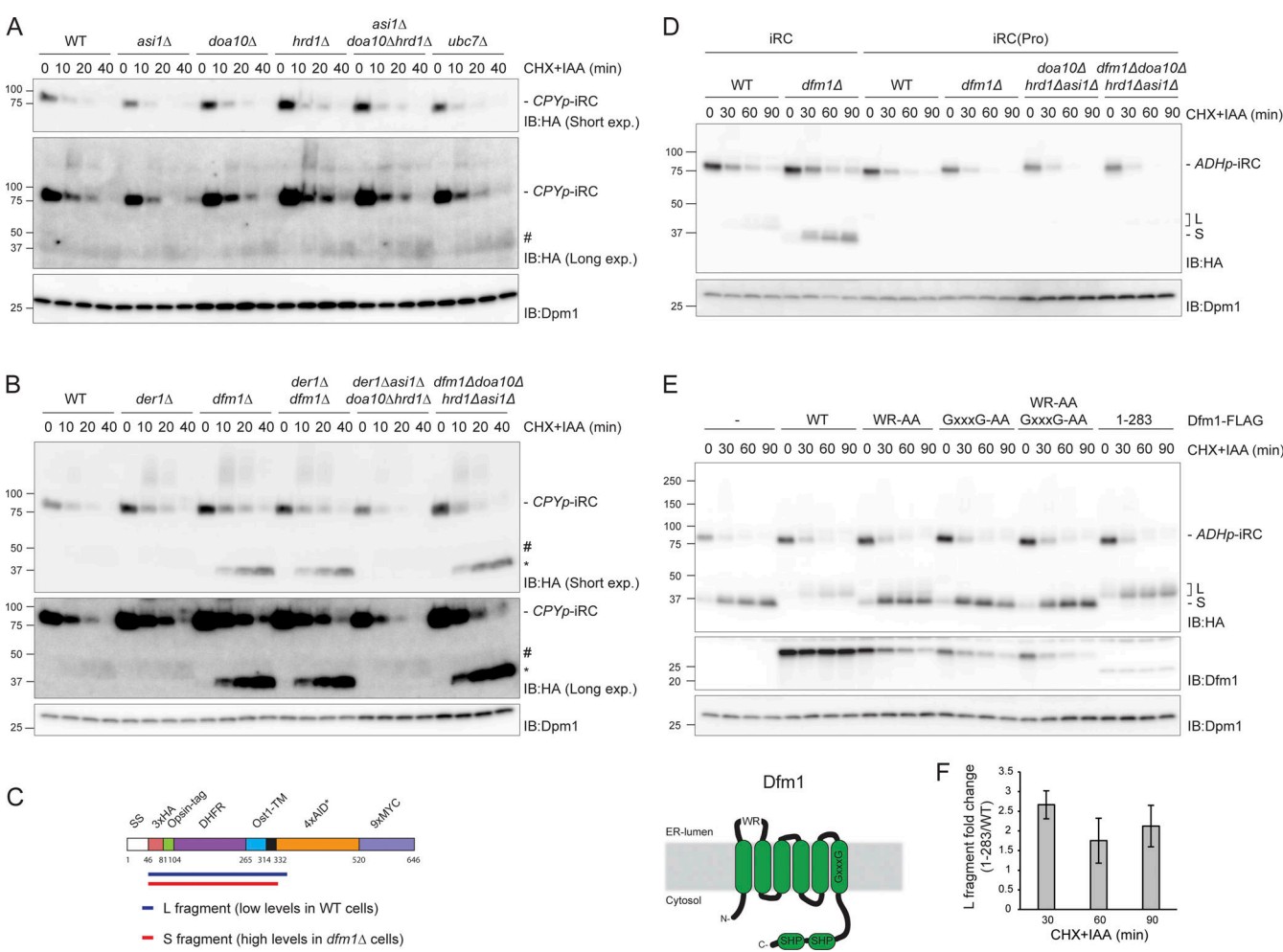

Figure 2. **Complete iRC degradation requires Dfm1. (A)** The degradation of iRC was analyzed in cells lacking the indicated ERAD components. Cells were treated with IAA and CHX and collected at the indicated time points. Whole-cell lysates were analyzed by SDS-PAGE and Western blot with the indicated antibodies. **(B)** The iRC degradation was analyzed as in A in WT cells or cells with the indicated deletions. Note that iRC degradation in cells lacking Dfm1 leads to the accumulation of a fragment of ~37 kD (*). This fragment is present in substantially higher amounts and is slightly smaller than the one observed in cells expressing Dfm1 (#). **(C)** Schematic representation of the iRC-derived L (long, in blue) and S (short, in red) fragments present in WT and *dfm1Δ* cells, respectively, as detected by mass spectrometry. Detergent extracts prepared from WT and *dfm1Δ* cells treated with auxin were subjected to immunoprecipitation with HA antibodies. Eluted proteins were analyzed by SDS-PAGE followed by mass spectrometry. **(D)** Degradation of iRC and its variant iRC(Pro), with three-point mutations that destabilize the tight folding of the DHFR domain, in WT cells or cells with the indicated deletions. Samples were prepared and analyzed as in A. **(E)** Top panel: The rhomboid and SHP motifs of Dfm1 are required for iRC degradation. *Dfm1Δ* cells with an empty vector (–) or the indicated FLAG-tagged *DFM1* variants were treated with IAA and CHX. Samples were collected at the indicated time points and analyzed as in A. Dfm1 was detected with α-Dfm1 antibody. Dpm1 was used as a loading control and detected with α-Dpm1 antibody. Bottom left panel: Schematic representation of Dfm1. WR and GxxxG motifs (Rhomboid-like features) and the Cdc48-interacting SHP boxes are depicted. **(F)** Quantification of the fold change of L fragment in cells expressing Dfm1[(1–283)]-FLAG mutant compared with WT. The graph shows the average of three independent experiments (*n* = 3), error bars represent the standard deviation. Source data are available for this figure: SourceData F2.

**Folded DHFR domain of iRC accumulates in Dfm1 mutant cells**

Derlins are small multipass ER membrane proteins implicated in the degradation of luminal (Knop et al., 1996; Lilley and Ploegh, 2004; Ye et al., 2004; Oda et al., 2006; Greenblatt et al., 2011) and membrane (Lilley and Ploegh, 2004; Ye et al., 2004; Stolz et al., 2010; Neal et al., 2018) ERAD substrates. They share similarities with rhomboid intramembrane proteases but lack catalytic activity (Greenblatt et al., 2011). and there are indications that they may be involved in retrotranslocation (Mehnert et al., 2014; Neal et al., 2018; Nejatfard et al., 2021; Wu et al., 2020). We tested the role of the yeast derlins Der1 and Dfm1 in iRC degradation. In the absence of Der1, Dfm1, or both, the degradation kinetics of full-

length iRC appeared unaffected (Fig. 2 B). Simultaneous deletion of Der1 or Dfm1 with all ERAD ubiquitin ligases also did not affect the turnover of full-length iRC. Strikingly, in cells lacking Dfm1, auxin-dependent degradation of iRC resulted in the prominent accumulation of a 37 kD fragment. This fragment was smaller than the one described earlier and appeared in much higher amounts (Fig. 2 B, compare # and *). Importantly, the generation of the shorter fragment was also inhibited if *dfm1Δ* cells were incubated with the proteasome inhibitor MG132 (Fig. S1 C). On the other hand, deletion of Pep4, essential for proteolytic activity in lysosomes, had no effect (Fig. S1 D). These data indicate that iRC degradation depends on the proteasome while lysosomal activity appears to be dispensable.

To determine the identity of the fragments generated in WT and *dfm1Δ* cells, they were immunoprecipitated and analyzed by mass spectrometry. Both the ~40 and 37 kD fragments, hereafter called L (long) and S (short) fragments, respectively, contain the luminal domains and transmembrane segment of the iRC, varying solely in the size of their cytosolic tails (Fig. 2 C). While the peptides identified from the S fragment contained 20 cytosolic residues adjacent to the TMD, the L fragment included an additional 18 amino acids toward the C-terminus encompassing the beginning of the AID* tag (Fig. 2 C and Fig. S1 E). Together, these data show that Dfm1 plays a unique and essential role in the degradation of the iRC luminal domain containing DHFR.

## Dfm1 requirement depends on the presence of iRC folded luminal domain

To test whether the accumulation of the S fragment was a direct consequence of DHFR's tight folding, we took advantage of DHFR(Pro), a mutant DHFR (Ala29Pro, Trp30Pro, Phe31Pro) with impaired folding stability (Bhamidipati et al., 2005). Upon addition of auxin, an iRC derivative with mutant DHFR(Pro), iRC(Pro) was degraded with faster kinetics to original iRC both in WT and *dfm1Δ* cells. The kinetics of iRC(Pro) degradation was also unaltered if mutations in Dfm1 and the ERAD ligases Asi1, Doa10, and Hrd1 were combined. Importantly, under all the conditions tested, iRC(Pro) was degraded entirely, without accumulation of any fragment, indicating that Dfm1 requirement increases for substrates with a tightly folded domain in the ER lumen (Fig. 2 D).

## Distinct Dfm1 domains are required for iRC degradation

Dfm1 consists of a rhomboid-like membrane domain followed by an extended cytosolic C-terminus encompassing two SHP boxes essential for the interaction with the Cdc48 ATPase complex (Fig. 2 E, bottom left panel) (Hitt and Wolf, 2004; Sato and Hampton, 2006; Greenblatt et al., 2011). Rhomboid-like signature motifs include the WR residues in a luminal loop and the GxxxG sequence in the sixth transmembrane segment, both required for the function of active rhomboid proteases as well as catalytically inactive derlins, such as Dfm1 (Lemberg et al., 2005; Greenblatt et al., 2011; Neal et al., 2018). We analyzed the contribution of these Dfm1 domains in iRC degradation. As expected, the expression of WT Dfm1 fused to a C-terminal FLAG tag reversed the accumulation of the S fragment observed in *dfm1Δ* cells. In contrast, individual and combined mutations in Dfm1 conserved rhomboid signature motifs did not (Fig. 2 E). The accumulation of iRC S fragment was observed even when Dfm1(WR-AA) or Dfm1(GxxxG-AA) were overexpressed, ruling out that the phenotype is due to the lower expression levels of these Dfm1 mutants (Fig. S2 A). Curiously, overexpression of Dfm1(WR-AA) in WT cells also resulted in the accumulation of the S fragment, suggesting a dominant-negative function of this mutation (Fig. S2 B). Thus, Dfm1 rhomboid-like activity is required for the degradation of the iRC-folded luminal domain.

To examine the role of Dfm1 SHP boxes in iRC degradation, we generated Dfm1(1–283), a C-terminal truncation that removes both SHP motifs that interact with Cdc48. Dfm1(1–283) did not affect the degradation rate of the full-length iRC; however, it led to a strong increase in the accumulation of the L fragment (Fig. 2, E and F). These data suggest that both Dfm1 rhomboid-like signature motifs and Cdc48-interacting SHP boxes contribute to iRC degradation.

## Cdc48 and its cofactors act at distinct steps of iRC degradation

The cytosolic Cdc48 ATPase, with the co-factors Npl4 and Ufd1, plays a central role in protein retrotranslocation from the ER (Wu and Rapoport, 2018). Since this Cdc48 complex is essential for cell viability, its involvement in retrotranslocation and degradation of iRC was tested in cells expressing temperature-sensitive (ts) alleles *cdc48-6*, with mutations in both D1 and D2 ATPase domains (Schuberth and Buchberger, 2005; Ruggiano et al., 2016), and *npl4-1*, which encodes a very unstable protein (DeHoratius and Silver, 1996). Both ts alleles retain residual Cdc48 activity at 25°C and are inactive at 37°C. Since *At*Tir1 is non-functional at 37°C (Nishimura et al., 2009), Tir1 from *Oryza sativa* (*Os*Tir1) was used to trigger iRC degradation in cells bearing ts alleles. While *Os*Tir1 is active at a wider temperature range, its auxin-dependent recognition of iRC is less stringent and basal degradation of iRC is observed even in the absence of auxin (Fig. S3 A).

Turnover of full-length iRC was unaffected in *cdc48-6* and *npl4-1* cells, irrespective of the growth temperature. However, in relation to WT cells, these mutants accumulated higher amounts of the L fragment, indicating that complete retrotranslocation and degradation of iRC requires a functional Cdc48 complex (Fig. 3 A and Fig. S3 B). Interestingly, complete inactivation of Cdc48 at 37°C resulted in L fragment–accumulation even in cells lacking Dfm1. However, if *cdc48-6 dfm1Δ* or *npl4-1 dfm1Δ* cells were grown at 25°C, there was an accumulation of the S fragment, likely due to the residual Cdc48/Npl4 activity. Therefore, Cdc48 and its cofactor Npl4 are necessary for the clearance of the iRC L fragment and its conversion into the S fragment in cells lacking Dfm1.

The involvement of the Cdc48 complex in iRC degradation was expected (Wu and Rapoport, 2018). However, the observation that Cdc48 inactivation did not affect the turnover of full-length iRC was surprising, particularly considering that inhibition of the proteasome, which functions at a downstream step, resulted in the stabilization of full-length iRC (Fig. 1 G and Fig. S1 C). Therefore, to dissect the role of Cdc48 in iRC degradation in further detail, we analyzed the contribution of additional Cdc48 adaptors and associated factors.

Other Cdc48 adaptors implicated in ERAD are Ubx2, which facilitates Cdc48 recruitment to the ER membrane (Schuberth and Buchberger, 2005; Neuber et al., 2005), and Ufd2, a ubiquitination enzyme that extends ubiquitin chains on substrates (Koegl et al., 1999; Rumpf and Jentsch, 2006). The degradation of iRC in *ubx2Δ* cells was indistinguishable from WT cells (Fig. S3 C). In contrast, cells lacking the polyubiquitination enzyme Ufd2 displayed a strong defect in iRC degradation, which accumulated primarily as the full-length protein and still localized at the ER (Fig. 3 B and Fig. S3 D). A similar phenotype was observed in cells lacking Rad23 and Dsk2 (Fig. 3 C), redundant shuttling factors recruited to the Cdc48 complex by Ufd2 and that facilitate the transfer of ubiquitinated substrates to the proteasome for degradation (Medicherla et al., 2004; Tsuchiya et al., 2017;

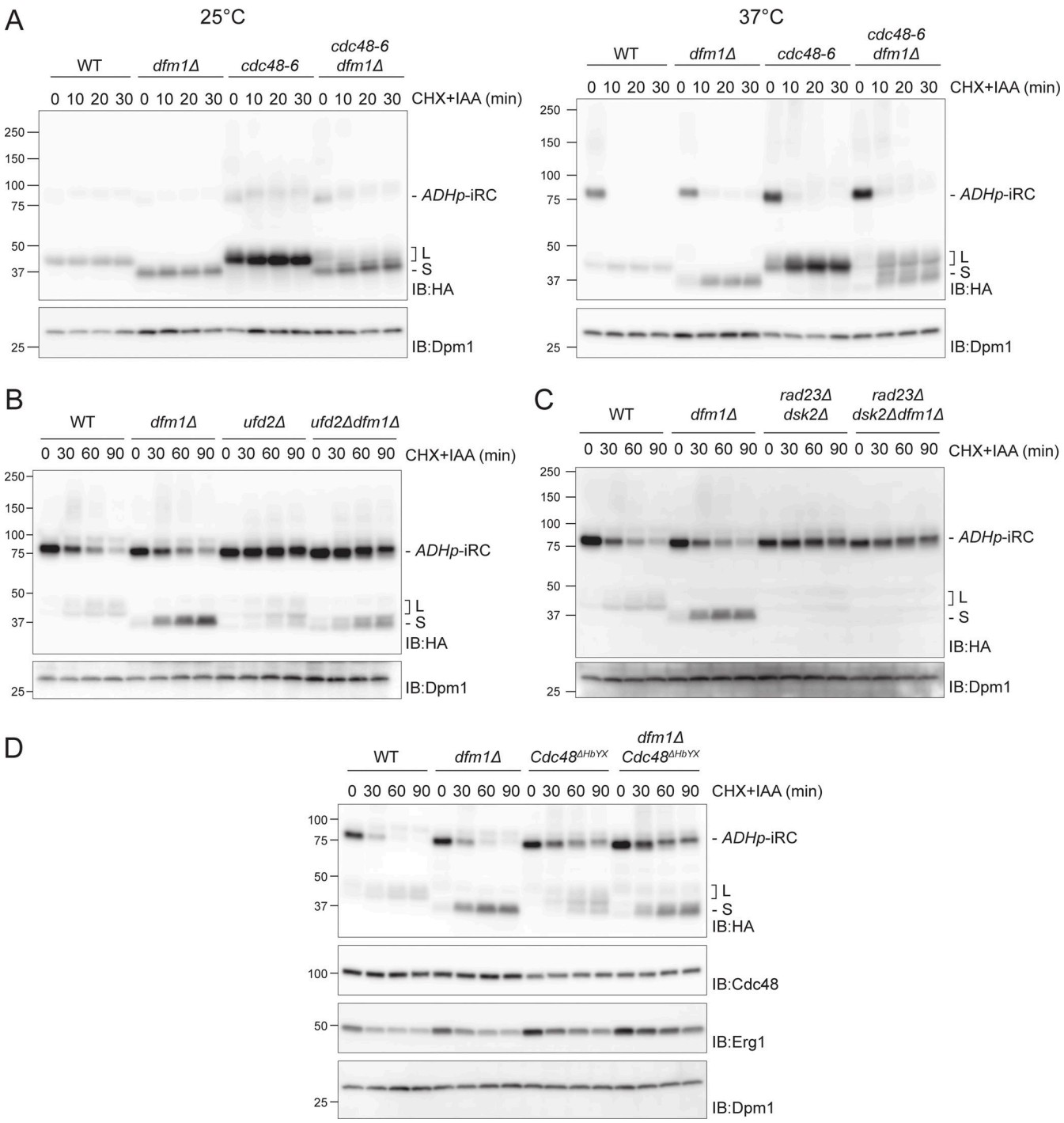

Figure 3. **Cdc48 and its cofactors act at distinct steps of iRC degradation. (A)** Time-course analysis of iRC degradation in cells with the indicated genotype at 25°C (left panel) or upon a 2 h shift to 37°C (right panel). At time 0, cells were treated with IAA and CHX. In these cells, iRC ubiquitination is mediated by *Os*Tir1, which is functional at 37°C. Whole-cell extracts were analyzed by SDS-PAGE (left panel: 4–20% gradient gel; right panel: 4–15% gradient gel) followed by Western blotting with α-HA antibody. Dpm1 was used as a loading control and detected with α-Dpm1 antibody. **(B and C)** Time-course analysis of iRC degradation in cells with the indicated genotype. Samples were prepared and analyzed as in Fig. 2 A. **(D)** Time-course analysis of iRC degradation in cells with the indicated genotype. Samples were prepared and analyzed as in Fig. 2 A. Cdc48 and the endogenous ERAD substrate Erg1 were detected with α-Cdc48 and α-Erg1 antibodies, respectively. Source data are available for this figure: SourceData F3.

Richly et al., 2005). If Ufd2 and Rad23/Dsk2 mutations were combined with deletion of Dfm1, mostly full-length iRC was still observed, indicating that Ufd2, Rad23, and Dsk2 act on iRC at an earlier step than Dfm1 (Fig. 3, B and C).

These results suggest that distinct steps in iRC degradation depend on different Cdc48 cofactors, with Ufd2 acting at an early step followed by Npl4/Ufd1. To test this possibility, we generated Cdc48$^{\Delta HbYX}$, a Cdc48 mutant lacking the last three C-terminal

amino acids (LYS) previously shown to be important for the interaction of Cdc48 or its mammalian homolog p97 with a subset of its cofactors (Allen et al., 2006; Zhao et al., 2007). Cdc48$^{\Delta HbYX}$ showed reduced binding to Ufd2 (Fig. S4 A), grew at a normal rate, and was competent in degrading the ERAD substrate CPY* (Fig. S4 B). Consistently, and like *ufd2Δ* and *rad23Δdsk2Δ* mutants, Cdc48$^{\Delta HbYX}$ cells accumulate mostly full-length iRC (Fig. 3 D). Together, these data indicate that Cdc48, with Ufd2 and the shuttling factors Rad23/Dsk2, act on full-length iRC while the clearance of iRC L fragment, containing the folded DHFR domain, depends on Npl4 and Cdc48 ATPase activity, as revealed by the *npl4-1* and *cdc48-6* ATPase deficient mutants.

### Cdc48 and Dfm1 interact with iRC during its degradation

Analysis of iRC degradation showed the requirement of Cdc48, its cofactors, and the ER membrane protein Dfm1. To investigate if and how these factors interacted with iRC during its degradation, we used immunoprecipitation. In WT cells, in the absence of auxin, iRC was a stable protein and its association with Cdc48 and Dfm1 was negligible (Fig. 4 A). Triggering iRC degradation by the addition of auxin resulted in robust association with both Cdc48 and Dfm1 (Fig. 4 A). Importantly, in cells expressing Cdc48$^{\Delta HbYX}$ the association of iRC with mutant Cdc48 and Dfm1 remained low after addition of auxin (Fig. 4 A). These results indicate that the recruitment of Cdc48 and Dfm1 to iRC depends on an intact Cdc48/Ufd2 complex. Indeed, the interaction between iRC and Cdc48 was also abrogated in *ufd2Δ* and *rad23Δdsk2Δ* mutants (Fig. 4 B). On the other hand, stimulation of iRC degradation in *dfm1Δ* cells reduced the association between iRC and Cdc48 suggesting that Dfm1, likely via its SHP boxes, stabilizes iRC association with Cdc48 complex (Fig. 4 A). Together, these data support a direct role of Cdc48 complex and Dfm1 in the degradation of iRC.

### Dfm1 couples retrotranslocation of the DHFR domain to its degradation

To better understand the roles of Dfm1 and Cdc48 in iRC retrotranslocation and degradation, we investigated the localization and topology of the L and S fragments. Fractionation experiments showed that both L and S fragments, generated in WT and *dfm1Δ* cells respectively, partition with cellular membranes (Fig. 5 A). This is consistent with the presence of the iRC transmembrane segment in both fragments (Fig. 2 C). To examine the topology of iRC fragments in relation to the ER membrane, we first analyzed their glycosylation status. To generate iRC fragments, WT and *dfm1Δ* cells were incubated with auxin. Subsequently, detergent-solubilized extracts from these cells were treated with Endoglycosidase H (Endo H) (Fig. 5 B) or PNGaseF (Fig. S5 A) and analyzed by SDS-PAGE and Western blotting. In WT and *dfm1Δ* cells, these treatments resulted in faster migration of full-length iRC, as expected. Similarly, Endo H or PNGaseF digestions of WT extracts resulted in faster migration of the L fragment, indicating that the N-terminal glycan was in the lumen of the ER, protected from endogenous peptide N-glycanase in the cytosol. In contrast, Endo H or PNGaseF treatment of extracts from *dfm1Δ* cells did not affect the mobility of the S fragment indicating that the

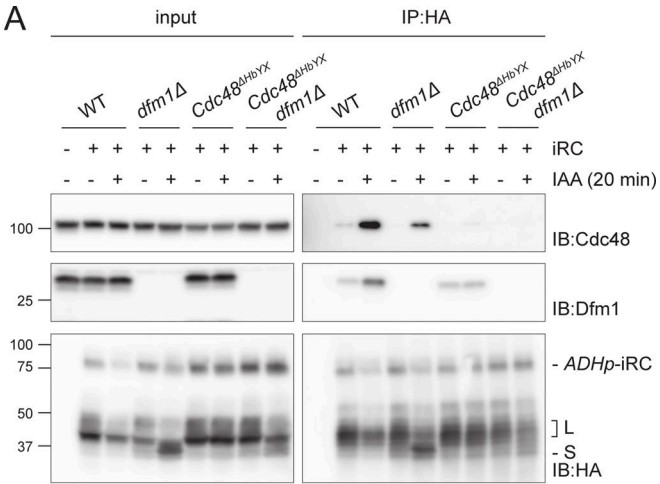

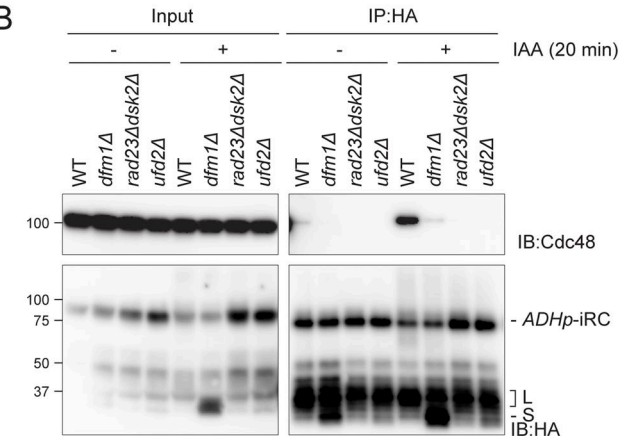

Figure 4. **Dfm1 and Cdc48 interact with iRC during its degradation.** **(A)** Cells with the indicated genotype were incubated for 20 min with or without IAA. Detergent solubilized membranes were subjected to immunoprecipitation with HA antibody, and the associated proteins were analyzed by SDS-PAGE and Western blotting. HA-tagged iRC was detected with the α-HA antibody. Cdc48 and Dfm1 proteins were detected with indicated antibodies. Note that in native extracts the iRC substrate is labile and the L fragment is generated even in the absence of auxin. **(B)** iRC interaction with Cdc48 ATPase is impaired in the indicated mutants. Samples were prepared and analyzed as in A. Source data are available for this figure: SourceData F4.

N-linked glycan proximal to its N-terminus had been exposed to and processed by endogenous peptide N-glycanase in the cytoplasm (Fig. 5 B and Fig. S5 A). Different topologies of iRC-derived L and S fragments were also supported by the proteinase K protection assay (Fig. S5 B). Together, these experiments indicate that the N-terminus of the iRC L fragment, detected at low levels in WT cells but abundant in Cdc48 ATPase-deficient mutants, was in the ER lumen. Moreover, the S fragment observed in *dfm1Δ* cells was, at least in part, exposed to the cytosol.

This was surprising and raised the possibility that, in *dfm1Δ* cells, iRC was retrotranslocated and that the S fragment was partitioned with membranes in the fractionation experiment simply because it contained the transmembrane segment (Fig. 2 C).

To test this hypothesis, we generated (Lum linker)-iRC, an iRC version with an extended unstructured 64 amino acid linker

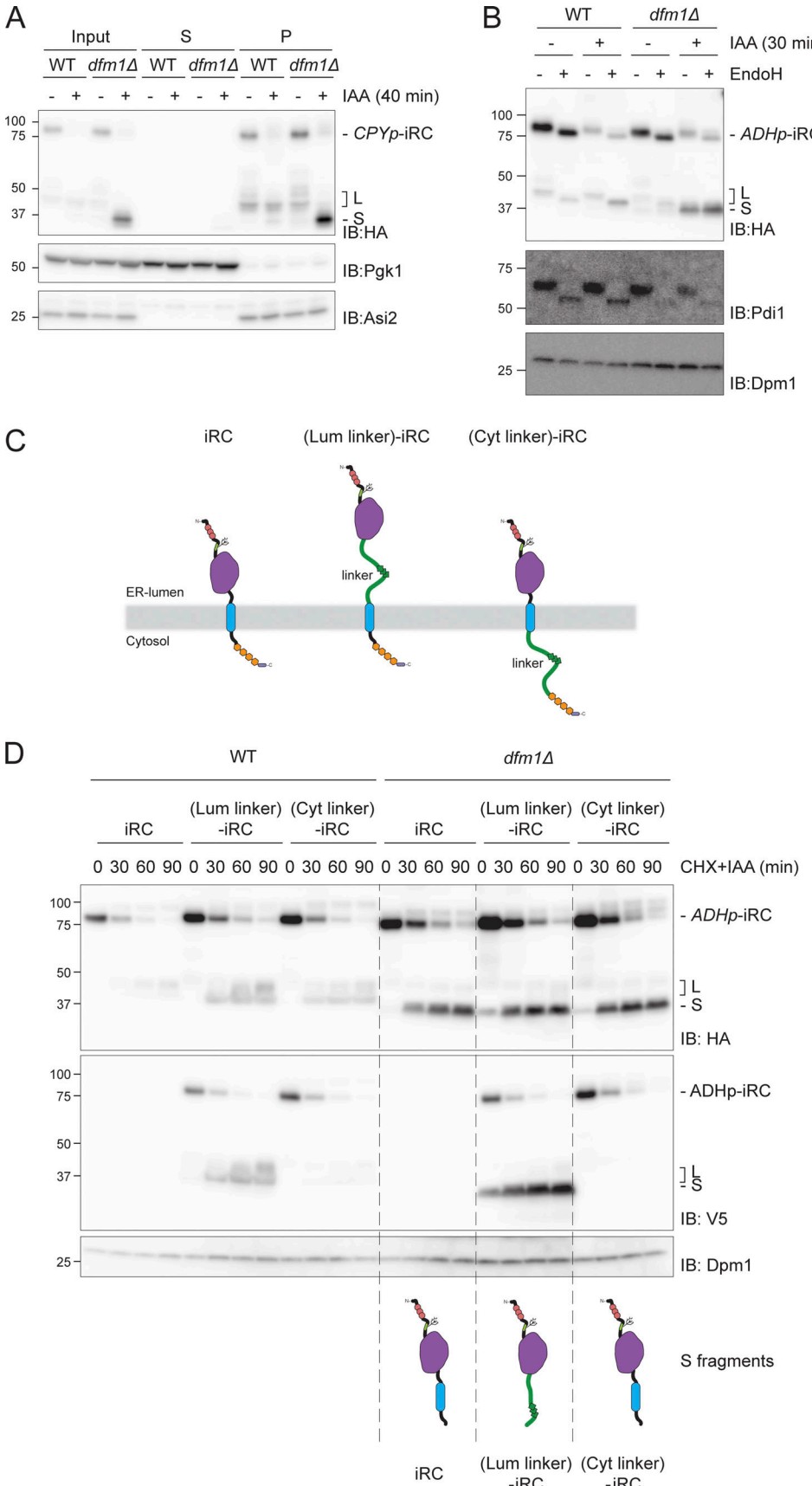

Figure 5. **iRC-derived L and S fragments have distinct topology. (A)** iRC-derived L and S fragments are membrane-associated. Cell extracts (Input) from WT and *dfm1Δ* cells were fractionated in crude membranes (P) and soluble (S) material. Where indicated, cells were incubated with IAA for 40 min. Fractions

were analyzed by SDS-PAGE and Western blotting using α-HA antibody. The ER membrane protein Asi2 and the cytosolic soluble Pgk1 were used as controls. Note that in native extracts, the iRC substrate is labile and the L fragment is generated even in the absence of auxin. **(B)** Glycosylation status of iRC fragments. Denaturing extracts from WT and *dfm1Δ* cells incubated with IAA for 30 min were prepared. Extracts were treated with Endo H and analyzed by SDS-PAGE and Western blotting using α-HA antibody. The glycosylated Protein Disulfide Isomerase 1 (Pdi1) was detected with α-Pdi1 antibody, while Dpm1 was used as loading control and detected with α-Dpm1 antibody. **(C)** Schematic representation of original iRC (left), (Lum linker)-iRC (middle), and (Cyt linker)-iRC (right). These latter substrates contain an 64 amino acid long linker (with a 3xV5 tag; shown in green) between DHFR and the transmembrane domain ([Lum linker]-iRC) or between the transmembrane domain and the AID motifs ([Cyt linker]-iRC). **(D)** Time-course analysis of the iRC-derivative substrates in WT and *dfm1Δ* cells. Samples were treated and analyzed as in Fig. 2 A. iRC derivatives were detected with α-HA and α-V5 antibodies. Schematic representation of the S fragments generated by iRC variants is shown at the bottom. In contrast to other iRC variants, the S fragment derived from (Lum linker)-iRC lacks the transmembrane segment (blue). Source data are available for this figure: SourceData F5.

(including 3xV5 epitope) in the ER lumen between the tightly folded DHFR domain and the transmembrane segment (Fig. 5 C). In parallel, a control substrate (Cyt linker)-iRC containing the same amino acid linker in the cytosolic region, between the TM and the AID* degron, was also generated (Fig. 5 C). The increased molecular weight resulting from the additional 64 residues in these iRC derivatives could not be resolved in the gradient gels used in most experiments (Fig. 5 D) but was clearly detected when the same samples were run in Tricine gels, which have higher resolution (Fig. S5 C). The derivatives were degraded upon the addition of auxin with kinetics similar to the original iRC and their degradation also resulted in the appearance of some L fragments in WT cells and abundant S fragments in *dfm1Δ* mutants (Fig. 5 D and Fig. S5 D). Importantly, in *dfm1Δ* cells, the S fragments generated from iRC or from its extended derivatives (Lum linker)-iRC and (Cyt linker)-iRC were of the same size, indicating that the degradation of all three substrates was blocked at the same point (Fig. 5 D and Fig. S5 C). Considering that the degradation of these proteins is initiated from their C terminus and that the S fragment observed for the various iRC derivatives is of the same size, we concluded that the transmembrane segment of (Lum linker)-iRC was degraded and was no more present in the resulting S fragment (Fig. 5 D). Importantly, Endo H glycan digestions performed as above showed similar topology for the fragments generated by iRC and (Lum linker)-iRC, with the L fragment being in the ER lumen and the S fragment being exposed to the cytosol (Fig. 6 A). This conclusion was further supported by proteinase K experiments (Fig. 6 B). The distinct topology of L and S fragments was also confirmed in *cdc48-6* mutant cells, which are defective in protein retrotranslocation upon a shift to the restrictive temperature. As observed with the original iRC (Fig. 3 A), the L fragment accumulated upon Cdc48-6 inactivation is protected upon proteinase K treatment (Fig. 6 B, green arrowhead). On the other hand, in *cdc48-6 dfm1Δ* cells, which accumulate both fragments (S fragment generated prior and L fragment after Cdc48-6 inactivation by temperature shift), only the S fragment was digested by proteinase K (Fig. 6 B, orange arrowhead). It should be noted that in cells subjected to a temperature shift to inactivate Cdc48-6, we observed partial digestion of the luminal protein in the absence of Triton X-100, suggesting a partial lysis of the isolated microsomes. We concluded that according to all criteria tested, (Lum linker)-iRC behaves similarly to iRC.

A key difference to iRC is that, in *dfm1Δ* cells, (Lum linker)-iRC generates an S fragment lacking transmembrane segment, simplifying its analysis by biochemical fractionation. Therefore,

using this approach, we observed that the L fragment derived from (Lum linker)-iRC was membrane-associated both in WT and *cdc48-6* cells, as expected (Fig. 6 C). In contrast, in *dfm1Δ* cells, the S fragment was partitioned almost exclusively to the soluble cytosolic fraction (Fig. 6 C). Importantly, the levels of cytosolic S fragment were reduced in *cdc48-6 dfm1Δ* cells, indicating that it corresponds to a retrotranslocated fragment. This conclusion was also supported by a cysteine accessibility assay, where intact membrane lysates were treated with the membrane-impermeable maleimide polyethylene-glycol (mal-PEG), which can covalently bind to two cysteines present in the DHFR module (Fig. S5 E). Altogether these data indicate that Cdc48, likely via ATP hydrolysis, can drive retrotranslocation of iRC, while Dfm1 ensures that iRC retrotranslocation and degradation are coupled.

## Discussion

Retrotranslocation of folded protein domains is challenging and their presence slows down the degradation of ERAD substrates (Tirosh et al., 2003; Bhamidipati et al., 2005; Shi et al., 2019). However, how proteins with folded luminal domains are retrotranslocated across the ER membrane and how this process is coordinated with downstream ERAD steps remains unclear. To investigate this issue, we developed iRC, a conditional membrane substrate whose biogenesis and folding can be uncoupled from its degradation, triggered by auxin.

In the absence of auxin, the iRC is a long-lived ER membrane protein consistent with it being properly folded and not recognized by quality control processes such as ERAD. The addition of auxin triggers iRC ubiquitination and ultimately its degradation by the proteasome. Our data indicate that iRC degradation is reminiscent of ERAD and involves several ERAD factors in the ER membrane and the cytosol. Analysis of iRC degradation in a variety of mutant backgrounds and conditions enabled the identification of distinct steps in iRC processing, all of which involved the AAA-ATPase Cdc48 and several of its cofactors (Fig. 7). Upon auxin addition, iRC ubiquitination led to a quick association of Cdc48. This initial step required the Cdc48 cofactor Ufd2, likely due to its ability to extend ubiquitin chains on substrates (Koegl et al., 1999; Rumpf and Jentsch, 2006). With the help of the shuttle factors Rad23/Dsk2 (Medicherla et al., 2004; Tsuchiya et al., 2017; Richly et al., 2005), Cdc48/Ufd2 facilitated the engagement of the proteasome and consequently in the degradation of the most C-terminal region of iRC in the cytosol. This step resulted in the generation of an iRC-derived

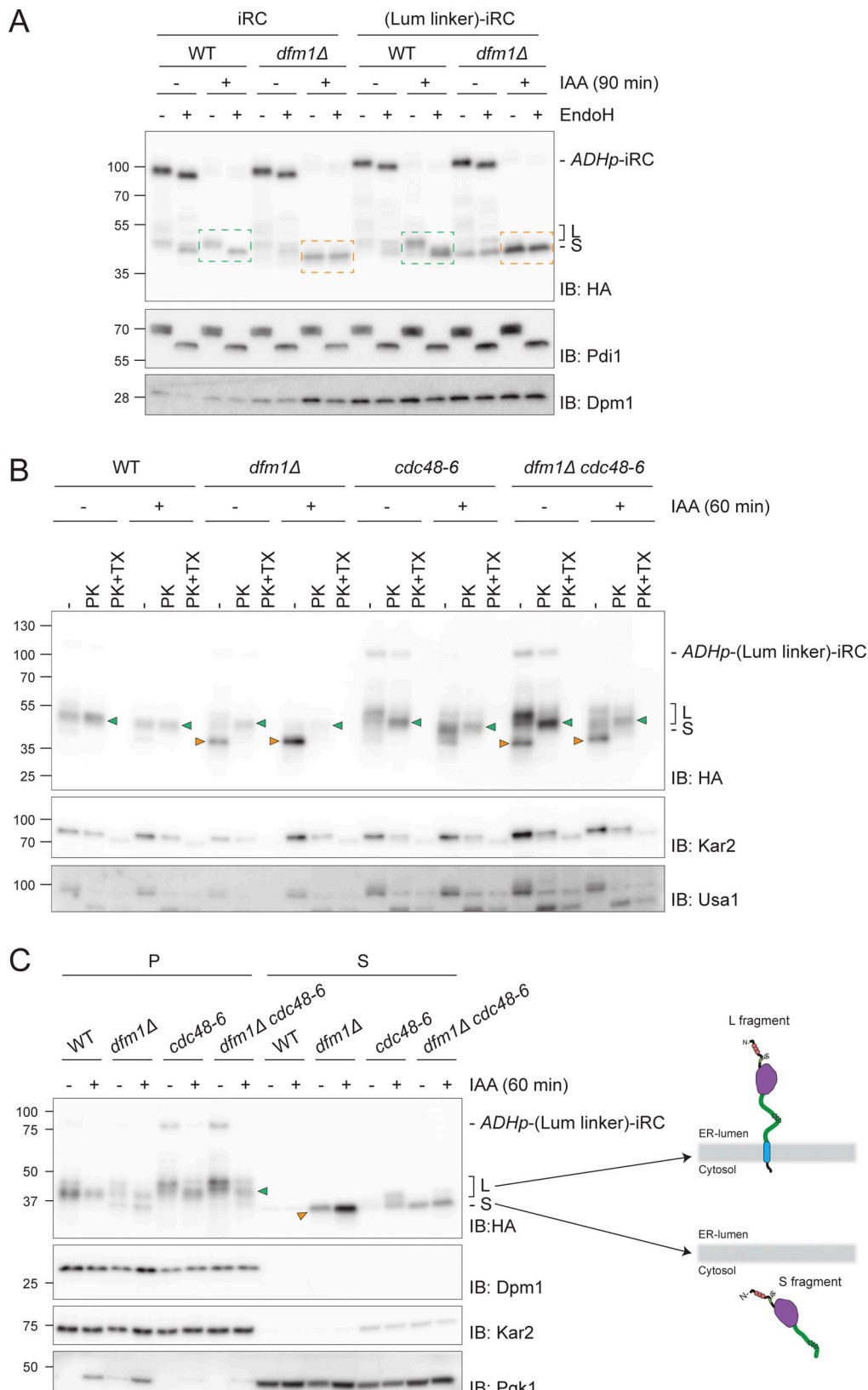

Figure 6. **Dfm1 couples retrotranslocation of the DHFR domain to its degradation. (A)** Analysis of glycosylation status of (Lum linker)-iRC as in Fig. 5 B. Green boxes: glycosylated L fragment; orange boxes: non-glycosylated S fragment. **(B)** Protease accessibility of (Lum linker)-iRC and derived fragments. Cells with the indicated genotype were shifted to 37°C for 2 h (to inactivate *cdc48-6*), IAA-treated (for 60 min), or left untreated and used to prepare ER-derived microsomes. These microsomes were incubated with Proteinase-K (PK) (50 µg/ml) in the presence or absence of 1% Triton X-100 (Tx). Samples were analyzed by SDS-PAGE followed by Western blotting using α-HA antibody. The ER luminal protein Kar2 and the ER membrane protein Usa1 were used as controls and

detected with α-Kar2 and α-Usa1 antibodies, respectively. Note that in native extracts the iRC substrate is labile and the L fragment is generated even in the absence of auxin. Green and orange arrowheads indicate L and S fragments, respectively. **(C)** In *dfm1Δ* cells, (Lum linker)-iRC-derived S fragment accumulates in the cytosol. Extracts from cells with the indicated genotype were fractionated in crude membranes (P) and soluble (S) material. Cells were shifted to 37°C for 2 h and, where indicated, were incubated with IAA for 60 min. Fractions were analyzed by SDS-PAGE and Western blotting using α-HA antibody. The ER membrane protein Dpm1, the ER luminal protein Kar2, and the cytosolic soluble Pgk1 were used as controls. On the right panel, schematic depiction of L and S fragments derived form (Lum linker)-iRC. Note that in native extracts the iRC substrate is labile and the L fragment is generated even in the absence of auxin. Green and orange arrowheads indicate L and S fragments, respectively. Source data are available for this figure: SourceData F6.

membrane-bound fragment containing the folded domain in the ER lumen (L fragment). Cells with mutations in Cdc48 ATPase domains or Npl4, a cofactor essential for substrate engagement, were unaffected in this initial step, suggesting that it does not involve ATP-dependent pulling on the substrate. On the other hand, these mutants accumulated large amounts of L fragment, indicating that the degradation of this remaining iRC fragment involves the ATPase activity of Cdc48 and its cofactors Npl4/Ufd1.

Auxin addition also resulted in the association of iRC with the derlin Dfm1. Importantly, Dfm1 was required for the degradation of iRC L fragment, with Dfm1 membrane rhomboid signature motif and the cytosolic SHP boxes playing distinct roles. Deletion of Dfm1 SHP boxes resulted in the accumulation of iRC L fragment, as was observed in Cdc48 mutants with impaired ATPase activity. This is in agreement with the role of Dfm1 SHP boxes in binding to Cdc48 to facilitate its engagement with ubiquitinated proteins in the ER membrane (Sato and Hampton, 2006; Neal et al., 2018).

In contrast, in *dfm1Δ* cells or expressing mutations in Dfm1 rhomboid signature motif, the degradation of iRC was blocked at a different step, resulting in the cytosolic accumulation of a smaller fragment containing the DHFR folded domain (S fragment). The accumulation of this cytosolic fragment in *dfm1Δ* cells was dependent on the presence of a luminal folded domain in the substrate, as iRC(Pro), an analogous substrate with a luminal unfolded DHFR domain, was degraded entirely in *dfm1Δ* cells. The accumulation of S fragment in the cytosol was also dependent on a functional Cdc48/Npl4/Ufd1 complex and was observed even upon deletion of all other ERAD membrane components previously implicated in protein retrotranslocation (Hrd1, Doa10, and Asi1). These data indicate that Cdc48 ATPase can drive retrotranslocation of the iRC luminal domain independently of Dfm1 and other canonical ERAD membrane factors.

These results were surprising considering that Dfm1 was previously implicated in the retrotranslocation of certain misfolded membrane proteins (Stolz et al., 2010; Neal et al., 2018,

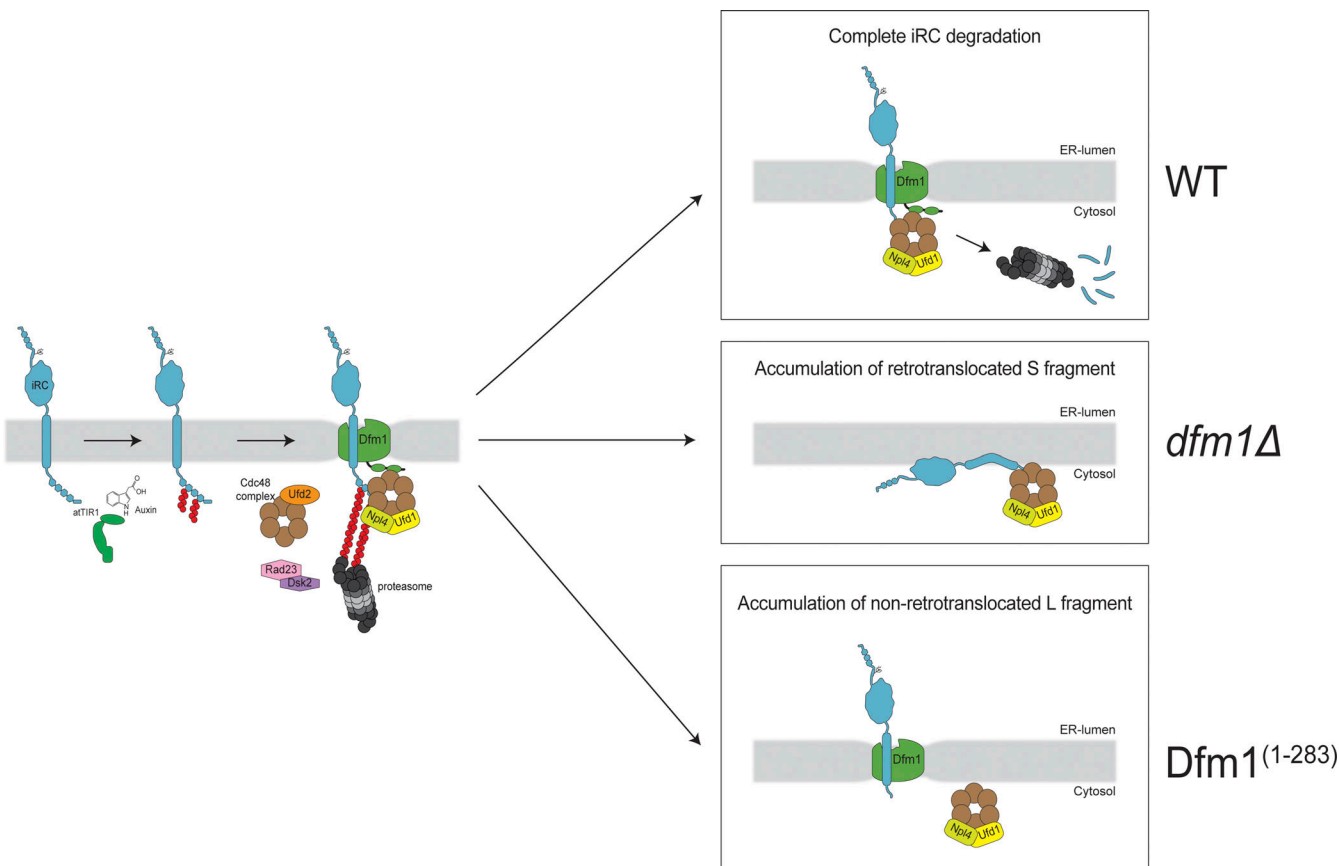

Figure 7. **An outline of the pathway for iRC degradation.** See Discussion for details.

2020), an activity that depended on Dfm1 rhomboid-like domain (Nejatfard et al., 2021). Like in other rhomboid proteins (Bondar et al., 2009; Engberg et al., 2022; Wang et al., 2006; Wu et al., 2006, 2020; Ben-Shem et al., 2007), the Dfm1 rhomboid-like domain appears to distort and thin the surrounding membrane, offering an appealing mechanism to move a polypeptide across the bilayer (Nejatfard et al., 2021). To cleave their substrates, active rhomboids also induce the local unfolding of their substrates within the membrane (Urban and Freeman, 2003; Akiyama and Maegawa, 2007). Possibly both membrane thinning and unfolding activities are employed by Dfm1 in ensuring the complete degradation of iRC. While membrane thinning may lower the energetic barrier that the Cdc48 pulling force needs to overcome, the destabilization of DHFR folding by Dfm1 may ensure that retrotranslocation occurs processively and is coupled to degradation by the proteasome. We cannot exclude that additional, unidentified factors work together with Dfm1 at this step. Although future experiments should test this model directly, it is consistent with the observation that *dfm1Δ* cells did not show any defect in the auxin-induced degradation of iRC(Pro) or the ERAD of Erg1, a Doa10 substrate (Foresti et al., 2013), that do not contain folded ER luminal domains.

In *dfm1Δ* cells, the Cdc48/Npl4/Ufd1 complex promotes the retrotranslocation of the iRC folded luminal domain, monitored in our assay by the L to S fragment conversion. However, this is inhibited in cells expressing Dfm1$^{(1–283)}$, which accumulate large amounts of the L fragment. This mutation disrupts Cdc48 interaction but does not appear to interfere with Dfm1 substrate binding (Neal et al., 2018; Nejatfard et al., 2021). It is unclear how Dfm1$^{(1–283)}$ inhibits retrotranslocation, but by binding to iRC, it may act as an intramembrane chaperone as recently proposed (Kandel et al., 2023). Alternatively, binding of iRC to Dfm1$^{(1–283)}$ may prevent its binding to other unknown factors that could aid in Cdc48/Npl4/Ufd1-dependent retrotranslocation in *dfm1Δ* cells. While future studies should further explore these possibilities, it appears clear that the Dfm1 rhomboid-like domain binding to iRC regulates its retrotranslocation.

In summary, here we delineated the steps leading to the degradation of iRC, a model membrane protein with folded luminal domain. While the iRC is useful for mechanistic studies, it will be of great interest to test whether endogenous membrane proteins with large luminal domains follow a similar degradation mechanism. Our data indicate that the pulling force generated by Cdc48 ATPase is sufficient to retrotranslocate a protein with a folded luminal domain. However, Dfm1 is essential to couple retrotranslocation to degradation, thereby preventing the accumulation of dead-end intermediates.

## Materials and methods
### Yeast strains and plasmids
The strains used are isogenic either to BY4741 (MATa *ura3Δ0 his3Δ1 leu2Δ0 met15Δ0*) or FY251 (MATa *ura3-52 his3Δ200 leu2Δ1 trp1Δ63*) and are listed in the Table S1. Tagging of proteins and individual gene deletions were performed by standard PCR-based homologous recombination (Longtine et al., 1998) or CRISPR (Laughery et al., 2015).

Strains with multiple gene deletions and temperature-sensitive alleles were made by crossing haploid cells of opposite mating types, followed by sporulation and tetrad dissection using standard protocols (Fink and Guthrie, 1991). All cloning procedures were carried out with PCR and Gibson-based assembly or restriction ligation. Dfm1-FLAG construct was generated by genomic amplification of *DFM1* gene and cloned into a pRS425-*GALp* with a single FLAG tag at the C-terminus. Dfm1 signature motif mutants were generated by site-directed mutagenesis. *DFM1* variants were subsequently subcloned into pRS415-*ADHp* for constitutive expression. Coding sequences from the iRC construct and *O. sativa TIR1* were ordered as gene blocks. For all strains, a minimal domain of IAA17 consisting of amino acids 71–114 as described in Morawska and Ulrich (2013) was used. Following its verification by sequencing, the iRC and variant constructs were amplified and inserted either with a CPY promoter or an ADH promoter into the *HO* locus of all strains of interest. Plasmids and primers used in this study are listed in Tables S2 and S3, respectively.

### Substrate degradation experiments
Cycloheximide shutoff chases were performed as described previously (Foresti et al., 2014). Briefly, yeast cells were grown either in rich media (YPD) or synthetic media with 2% glucose in the exponential phase (OD$_{600}$ 0.6–0.9) at 25°C. Overexpression of Dfm1-FLAG variants was driven from the inducible GAL promoter upon constant growth in synthetic media (SC-Leucine) with 4% D-galactose to the exponential phase at 25°C. Six ODs of cells were harvested, centrifuged for 5 min at 3,000 *g*, and resuspended in the appropriate media at 1 OD/ml. For auxin-dependent degradation experiments, auxin (Indole-3-acetic acid, IAA) was added to the cultures from a stock prepared in DMSO to a final concentration of 400 μM, unless stated differently. In experiments with cycloheximide, the drug was used at 250 μg/ml from a stock prepared in H$_2$O. For proteasome-dependence experiments, cells were treated with the proteasome inhibitor MG132 (final concentration 25 μg/ml in DMSO) 30 min before the addition of cycloheximide and IAA. At the specified time points, 1 OD of cells was collected and whole-cell lysates were prepared using NaOH extraction. Cell pellets were treated with 300 μl of 150 mM NaOH for 10 min on ice, briefly centrifuged, and resuspended in 65 μl sample buffer (SB; 100 mM Tris/HCl pH 6.8, 3% SDS, 15% glycerol, 50 mM DTT). Samples were then analyzed by SDS-PAGE (0.2–0.4 ODs per sample were loaded) followed by immunoblot with the indicated antibodies. Unless indicated, chases were performed at 25°C. Temperature-sensitive mutants were grown at 25°C and shifted to 37°C for 1–2 h prior to the addition of IAA.

### Western blotting
Protein samples were analyzed by SDS-PAGE on 4–20% gradient Criterion TGX precast gels (Bio-Rad), transferred onto a PVDF membrane, and incubated with the indicated antibodies. The signal was analyzed by enhanced chemiluminescence (Western Lightning ECL Pro, Perkin Elmer), visualized with Amersham Imager 600 (GE Healthcare Life Sciences), and quantified using

Image Studio software (Li-Cor). Representative images of three independent experiments are shown.

For Tris/Tricine gel, a self-casted 10% acrylamide gel was run in a Tris/Tricine running buffer (running buffer: 0.1 M Tris/HCl pH 8.3, 0.1 M Tricine, 0.1% SDS; anode buffer: 0.2 M Tris/HCl pH 8.8). The gel was transferred and analyzed as described above.

Antibodies used in this study are listed in Table S4.

## Glycosylation assay

For PNGase treatment cell lysates obtained by NaOH extraction were diluted 1:1 in water to the final concentration of SB. PNGase F (0.5 µl, 2,500 U) was added to 20 µl of lysate and incubated for 15 min at 37°C. Samples were then analyzed by SDS-PAGE followed by immunoblot.

For Endo H treatment, 30 ODs of cells were lysed in 1 ml of Lysis Buffer (LB; 50 mM Tris/HCl pH7.4, 200 mM NaCl, 1 mM EDTA, 2 mM phenylmethylsulfonyl fluoride (PMSF) and protease inhibitor cocktail) by 6 × 30 sec cycles of bead-beating at 4°C. Lysates were cleared by a 15-minute centrifugation at 300 $g$. Cell lysates were denatured with Protein Denaturing Buffer (0.5% SDS, 40 mM DTT) at 95°C for 10 min. Subsequently, 500 U of Endo H (New England Biolabs) were added with the supplier buffer (50 mM sodium acetate pH 6) and incubated at 37°C for 2 h. Samples were diluted 1:1 in SB and analyzed by SDS-PAGE and immunoblot.

## Membrane fractionation experiments

For cell fractionation experiments, 50 ODs of cells were lysed in LB buffer by 6 × 30 s cycles of bead-beating at 4°C. Lysates were cleared by a 10-min centrifugation at 300 $g$. Crude membrane and soluble fractions were obtained from cleared lysates by centrifugation at 100,000 $g$ for 25 min at 4°C. The membrane pellet was resuspended in SB and solubilized at 65°C for 10 min. The cell lyses and the soluble fractions were diluted 1:1 in SB buffer. Samples were analyzed by SDS-PAGE and immunoblot.

## Coimmunoprecipitation

For coimmunoprecipitation of iRC in *dfm1Δ* and Cdc48$^{ΔHbYX}$ cells, 100 ODs were harvested and lysed as described for membrane fractionation experiments. Cell lysates were solubilized in LB + 1% DMNG (Decyl Maltose Neopentyl Glycol) supplemented with 1 mM PMSF and protease inhibitor cocktail. After 2 h, the solubilized lysates were precleared by centrifugation for 20 min at max speed on a table-top centrifuge and the supernatant was incubated overnight with 30 µl of anti-HA magnetic beads (Pierce TM). Beads were washed three times with LB + 1% DMNG, and bound proteins were eluted with SB buffer and analyzed by SDS-PAGE and immunoblotting.

For coimmunoprecipitation of iRC in *dfm1Δ*, *ufd2Δ*, and *rad23Δdsk2Δ*, 100 ODs of cells were harvested and lysed as described for membrane fractionation experiments, except that the membrane fraction was solubilized in LB + 1% GDN (glycodiosgenin) supplemented with 1 mM PMSF and protease inhibitor cocktail. After 2 h, the solubilized membranes were precleared by centrifugation for 15 min at max speed on a table-top centrifuge and the supernatant was incubated overnight with 20 µl of anti-HA magnetic beads (Pierce TM). Beads were washed three times with LB + 0.02% GDN, and the bound proteins were eluted with SB buffer and analyzed by SDS-PAGE and immunoblotting.

For coimmunoprecipitation of Cdc48 with its cofactors, 50 ODs were lysed as described above and solubilized with 1% DMNG for 2 h. The solubilized lysates were precleared by centrifugation for 20 min at max speed on a table-top centrifuge and incubated with α-Cdc48 antibody or IgG isotype for 2 h. The samples were then incubated for 2 h with 20 µl of Protein A magnetic beads. Beads were washed three times with LB + 1% DMNG, and bound proteins were eluted with SB buffer and analyzed by SDS-PAGE and immunoblotting.

## Protease protection assay

100 ODs of cells were harvested, resuspended in resuspension buffer (100 mM Tris/HCl pH 9.5, 10 mM DTT), and incubated at 25°C for 10 min. The cells were recovered (1,000 g, 5 min), resuspended in spheroplasting buffer (10 mM Tris/HCl pH 7.4, 700 mM Sorbitol, 0.5% glucose, 1% yeast extract, 2% bactopeptone, 1 mM PMSF, and protease inhibitor cocktail), and incubated with zymolyase 20T (10 µg/OD) at 30°C for 30 min. Spheroplasts were recovered by centrifugation at 1,000 $g$ for 5 min, at 4°C, washed once with cracking buffer (50 mM Tris/HCl pH 7.4, 100 mM NaCl, 700 mM sorbitol, 1 mM EDTA), and lysed by douncing. The lysate was clarified (1,000 g, 3 min, 4°C) and incubated with proteinase K (50 µg/ml) or proteinase K with 1% Triton X-100 for 30 min on ice. The reaction was stopped by the addition of PMSF (4 mM final concentration) for 10 min on ice. The samples were diluted 1:1 in SB and analyzed by SDS-PAGE and immunoblotting.

## Cysteine accessibility assay

200 ODs of cells were harvested and lysed as described above for the protein protection assay. The lysates were incubated with maleimide–polyethylene-glycol (mal-PEG) (5 kD) (2.5 mM) with or without 1% Triton X-100 for 1 h on ice. The reaction was quenched by the addition of DTT (25 mM) for 30 min on ice. Subsequently, the samples were solubilized in LB + 1% Triton X-100 supplemented with 1 mM PMSF and protease inhibitor cocktail. After 1 h of rotating at 4°C, the solubilized lysates were cleared by centrifugation for 20 min at max speed on a table-top centrifuge at 4°C and the supernatant was incubated 2 h with 20 µl of anti-HA magnetic beads (Pierce TM). Beads were washed three times with LB + 1% Triton X-100 and the bound proteins were eluted with SB buffer and analyzed by SDS-PAGE and immunoblotting.

## Fluorescence microscopy

Cells were grown to OD = 1, treated as indicated, spun down, and resuspended in a synthetic medium for microscopy.

Fluorescence microscopy was performed at room temperature using the Zeiss Axio Observer.Z1 equipped with a Hamamatsu Orca Flash 4.0 digital CMOS camera and controlled by 3i Slidebook 6.0 software. Images were acquired with a Plan-Apochromat 100× (N.A. 1.4) objective using the GFP filter. Images were analyzed with 3i Slidebook 6.0 software.

## Mass spectrometry

Cells were grown in YPD until OD of 0.75 and added 0.5 mM auxin for 40 min. 150 ODs of cells were harvested and

membrane fraction was prepared using bead beating as described for membrane fractionation. The pelleted membranes were solubilized in 1.2 ml in LB supplemented with 1% Triton X-100 and incubated overnight with 75 µl anti-HA beads (Pierce TM). Beads were washed three times with LB + 1% Triton X-100 and the bound proteins were eluted with 100 µl SB buffer and 95% was separated by SDS-PAGE and Coomassie-stained. In parallel, immunoblotting was performed on 5% of the same sample against HA epitope. The fragment bands were excised from the Coomassie-stained gel and subjected to in-gel trypsin digestion as previously described (Shevchenko et al., 2006). After digestion, peptides were separated by nanoflow reversed-phase liquid chromatography coupled to Q Exactive Hybrid Quadrupole-Orbitrap mass spectrometer (Thermo Fisher Scientific). Peptides were loaded on a C18 PepMap100 pre-column (inner diameter 300 µm×5 mm, 3 µm C18 beads; Thermo Fisher Scientific) and separated on a 50-cm reversed-phase C18 column (inner diameter 75 µm, 2 µm C18 beads) using a linear gradient from 10 to 35% of B for 30 min at a flow rate of 200 nl/min (A: 0.1% formic acid, B: 0.1% formic acid in acetonitrile). All data were acquired in a data-dependent mode, automatically switching from MS to collision-induced dissociation MS/MS on the top 10 most abundant ions with a precursor scan range of 350–2,000 m/z. MS spectra were acquired at a resolution of 70,000 and MS/MS scans at 17,500. Dynamic exclusion was enabled with an exclusion duration of 5 s and charge exclusion was applied to unassigned and mono-charged ions. Raw data files were processed for protein identification using MaxQuant, version 1.6.3.4 (Tyanova et al., 2016). The MS/MS spectra were searched against the relevant Uniprot proteome database; precursor mass tolerance was set to 20 ppm and MS/MS tolerance to 0.05 Da. Enzyme specificity was set to trypsin with a maximum of two missed cleavages. The false discovery rate for protein and peptide spectral matches was set at 0.01.

### Online supplemental material

Fig. S1 shows iRC degradation in an auxin-dependent manner, the effect of proteasome inhibition on L and S fragments, and their mass spectrometry analysis. Fig. S2 shows that Dfm1 rhomboid motifs are required for iRC degradation. Fig. S3 shows that Cdc48 and its cofactors act at distinct steps of iRC degradation. Fig. S4 shows that the Cdc48$^{HbYX}$ mutant is defective in its interaction with Ufd2, but still functional. Fig. S5 shows the analysis of fragments topology. Table S1 lists the yeast strains used in this study. Table S2 lists the plasmids used in this study. Table S3 lists the oligonucleotides used in this study. Table S4 lists the antibodies used in this study.

### Data availability

All the data are included in the article or supplemental material.

## Acknowledgments

We thank M. Glickman (Technion, Haifa, Israel), S. Jentsch (deceased, formerly at MPI-Martinsried, Martinsried, Germany), F. Wilfing (MPI-Frankfurt, Frankfurt, Germany), and D. Wolf (deceased, formerly at University of Stuttgart, Stuttgart, Germany) for

reagents, members of the Carvalho lab for discussions, and R. Pisa, T. Rapoport, and A. Stein for comments on the manuscript. Proteomics experiments were performed at the Advanced Proteomics Facility, Department of Biochemistry, University of Oxford.

P. Carvalho was supported by an investigator award from The Wellcome Trust (223153/Z/21/Z) and a consolidator award from the European Research Council (grant #817708). Open Access funding provided by University of Oxford.

Author contributions: D.G. Vitali, D. Fonseca, and P. Carvalho performed the experiments; D.G. Vitali, D. Fonseca, and P. Carvalho conceived the study and analyzed the data. D.G. Vitali and P. Carvalho curated the data presentation and wrote the manuscript. P. Carvalho supervised and acquired the funding supporting the project.

Disclosures: The authors declare no competing interests exist.

Submitted: 14 August 2023

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

# Supplemental material

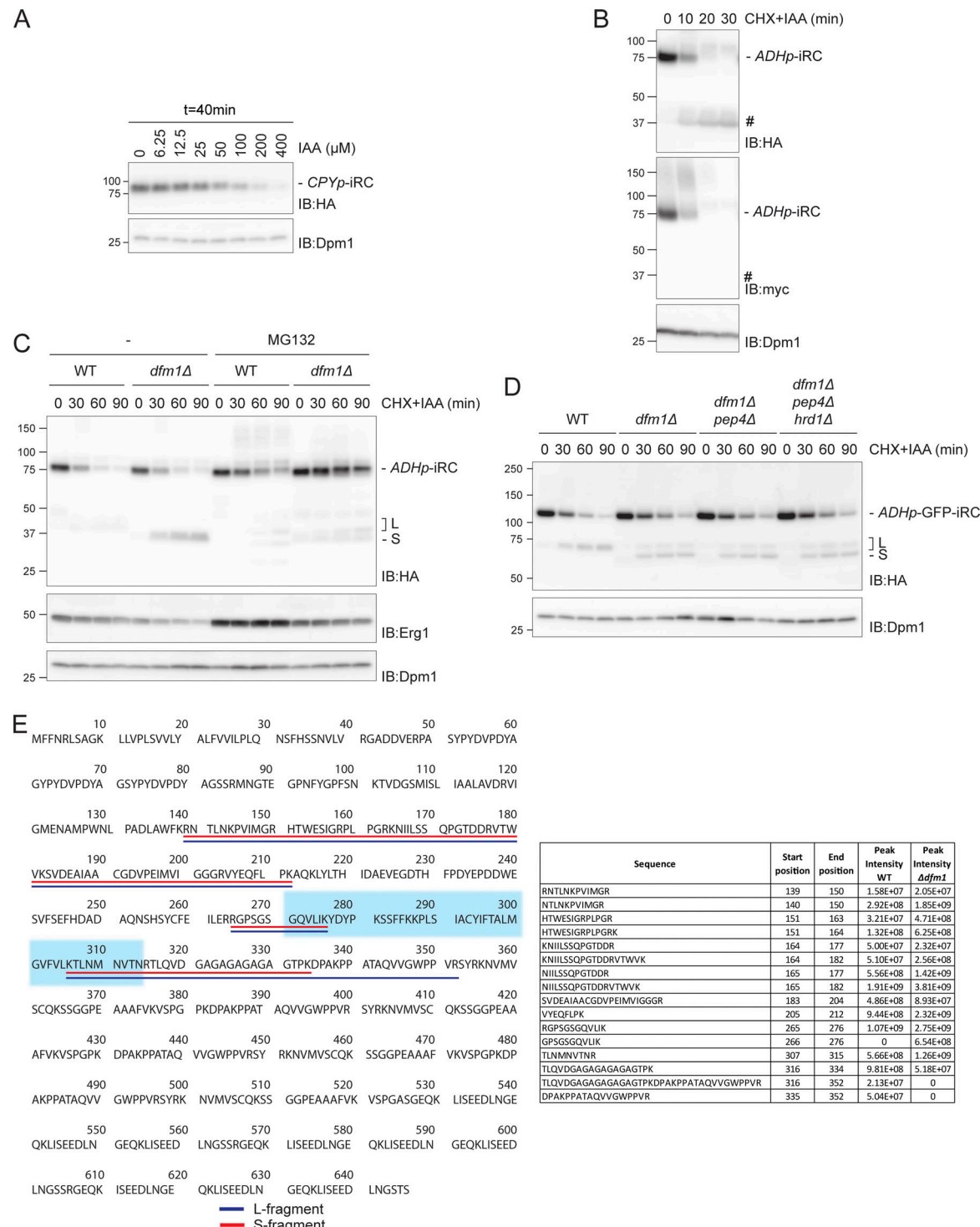

**Figure S1.  iRC is degraded vectorially from the C-terminus and L and S fragments generation requires the proteasome. (A)** iRC degradation depends on auxin concentration. Levels of iRC in WT cells expressing *At*Tir1 were analyzed after 40 min incubation with the indicated concentration of auxin (IAA). Samples prepared and analyzed as in Fig. 1 F. **(B)** Time-course analysis of iRC. Samples were prepared and analyzed as in Fig. 2 A. The substrate and the derived fragment were detected with α-HA and α-myc antibodies. To note, these cells expressed an untagged version of *At*Tir1, which leads to faster degradation of iRC. **(C)** Effect of proteasome inhibition on the degradation of iRC in *dfm1Δ* cells. Cells with the indicated genotype were treated with the proteasome inhibitor MG132 (25 µg/ml) for 30 min, and subsequently with IAA and CHX to follow iRC degradation. *PDR5* was deleted in *WT* and *dfm1Δ* cells to increase the effectiveness of MG132 treatment. Samples were analyzed as in Fig. 1 G. **(D)** Lysosome activity is not involved in iRC degradation. Samples were prepared and analyzed as in Fig. 2 A. The substrate and the derived fragment were detected with α-HA. Dpm1 was used as the loading control. **(E)** Left panel: amino acid sequence of iRC. Underlined in blue and red are the peptides identified by mass spectrometry for the L and S fragments, respectively. iRC transmembrane segment is highlighted in a light blue background. Right panel: the intensity of the peptides from L and S fragments in WT and *dfm1Δ* cells, respectively, as determined by mass spectrometry. Source data are available for this figure: SourceData FS1.

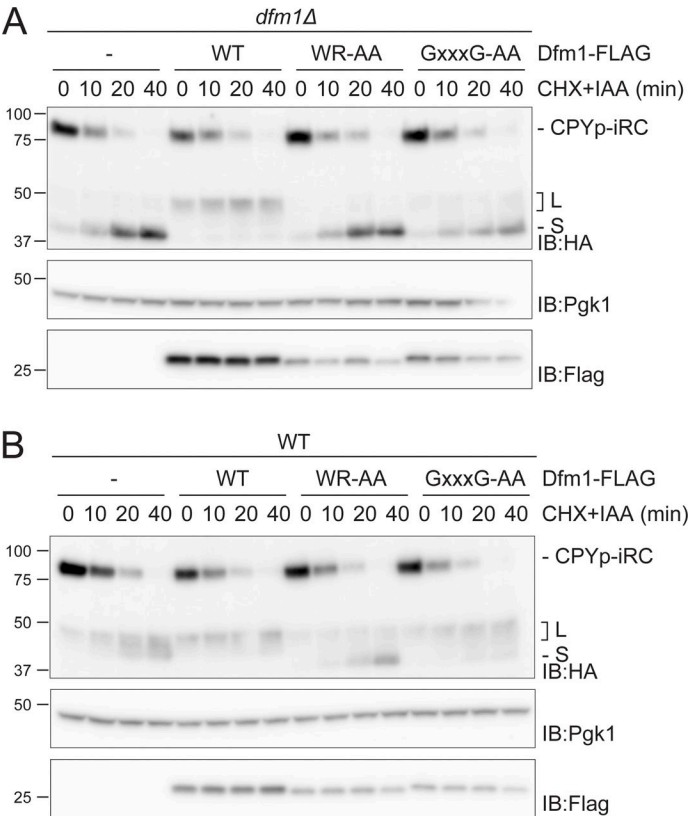

Figure S2. **Dfm1 rhomboid motifs are required for iRC degradation. (A)** Time-course analysis of iRC degradation in *dfm1Δ* cells with an empty vector (–) or overexpressing the indicated *DFM1* variants. Expression of Dfm1 variants was driven from the inducible GAL promoter upon growth in galactose-containing media. Samples were treated and analyzed as in Fig. 2 A. iRC and FLAG-tagged Dfm1 derivatives were detected with α-HA and α-Flag antibodies, respectively. Pgk1 was used as a loading control and detected with α-Pgk1 antibody. **(B)** Time-course analysis of iRC degradation in WT cells with an empty vector (–) or overexpressing the indicated *DFM1* variants. Samples were prepared and analyzed as in A. Source data are available for this figure: SourceData FS2.

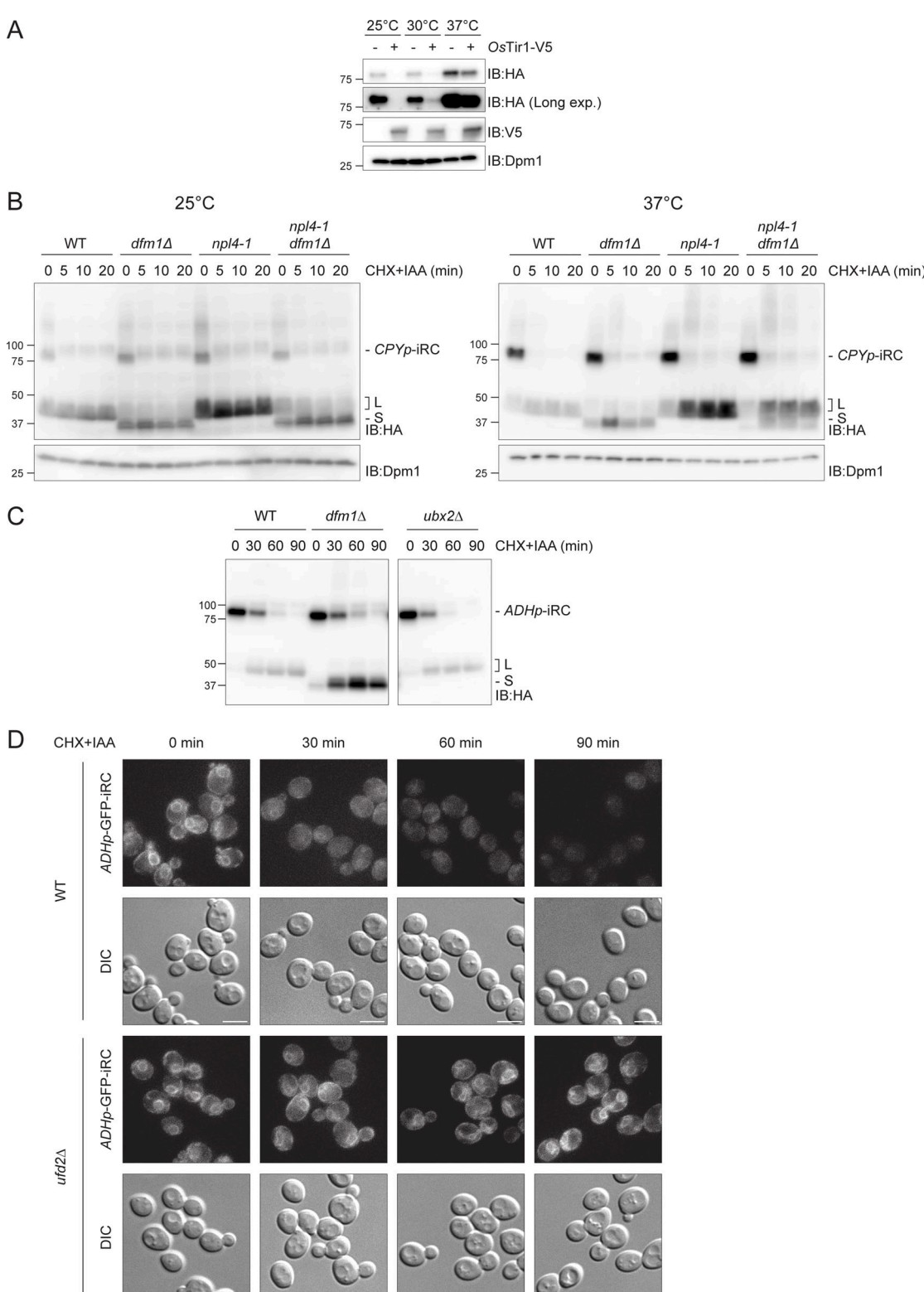

**Figure S3.** **Cdc48 and its cofactors act at distinct steps of iRC degradation. (A)** *Os*Tir1 promotes iRC degradation even in the absence of Auxin. WT cells expressing the iRC with and without V5-tagged *Os*Tir1 (*Os*Tir1-V5) were grown in the absence of auxin at the indicated temperature. Whole-cell lysates were analyzed by SDS-PAGE and immunoblotting with the indicated antibodies. Dpm1 was used as the loading control. **(B)** Time-course analysis of iRC degradation in cells with the indicated genotype at 25°C (left panel) or upon a 1 h shift to 37°C (right panel). At time 0, cells were treated with IAA (50 µM) and CHX. In these cells, iRC ubiquitination is mediated by *Os*Tir1, which is functional at 37°C. Samples were prepared and analyzed as in Fig. 2 A. Dpm1 served as loading control. **(C)** Time-course analysis of iRC degradation in cells with the indicated genotype. Samples were prepared and analyzed as in Fig. 2 A. **(D)** Fluorescence images of a time-course to follow the localization of GFP-tagged iRC in WT and *ufd2Δ* upon treatment with IAA and CHX. Scale bar: 5 µm. Source data are available for this figure: SourceData FS3.

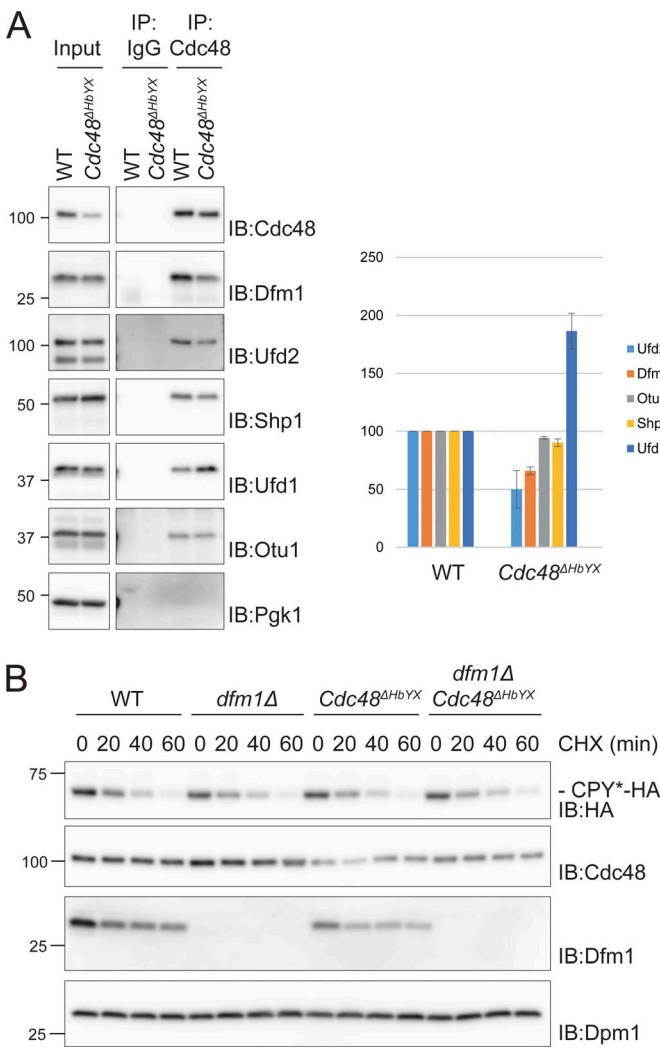

Figure S4. **Cdc48$^{\Delta HbYX}$ mutant is defective in its interaction with Ufd2 but still functional. (A)** Detergent-solubilized lysates from WT and Cdc48$^{\Delta HbYX}$ cells were subjected to immunoprecipitation with Cdc48 antibody, and associated proteins were analyzed by SDS-PAGE and Western blotting with indicated antibodies. Right panel: The amount of coprecipitating Cdc48 cofactors was quantified. Data were normalized for Cdc48 levels, and two independent experiments were quantified ($n$ = 2). Error bars represent standard deviation. **(B)** CPY* degradation monitored by CHX chase. The indicated cells with a plasmid expressing CPY*-HA were treated with CHX. Samples were collected at the indicated time points, and cell lysates were analyzed by SDS-PAGE and Western blot. CPY*-HA was detected with α-HA antibody. The control proteins Cdc48, Dfm1, and Dpm1 were detected with the respective antibody. Source data are available for this figure: SourceData FS4.

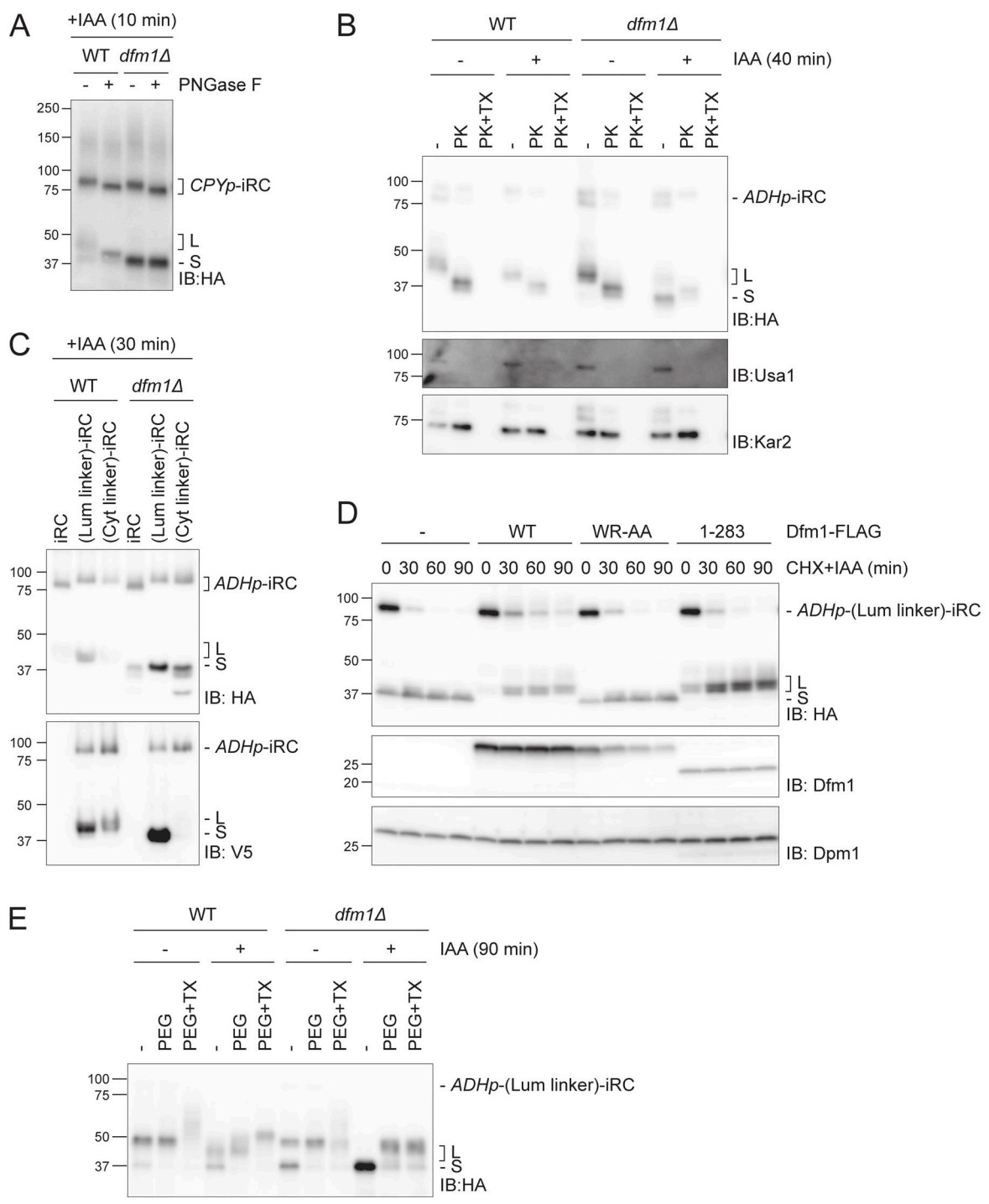

Figure S5. **Analysis of fragments topology. (A)** Glycosylation status of iRC fragments. Whole-cell extracts of WT and *dfm1Δ* cells expressing iRC and treated with IAA for 10 min were incubated with PNGase F and analyzed by SDS-PAGE and Western blotting using an α-HA antibody. **(B)** Protease accessibility of iRC and derived fragments. ER-derived microsomes from untreated or IAA-treated (for 40 min) WT and *dfm1Δ* cells were incubated with Proteinase-K (PK) (50 µg/ml) in the presence or absence of 1% Triton X-100 (Tx). Samples were analyzed by SDS-PAGE followed by Western blotting using α-HA antibody. The ER luminal protein Kar2 and the ER membrane protein Usa1 were used as controls and detected with α-Kar2 and α-Usa1 antibodies, respectively. **(C)** Samples from Fig. 5 D (*t* = 30 min) were analyzed on a 10% Tris-Tricine gel which gives better resolution. In this experiment, the molecular weight difference from the additional 64 amino acids in (Lum linker)-iRC and (Cyt linker)-iRC results in slower migration than original iRC. In contrast, the S fragments from all iRC derivatives show the same molecular weight indicating their degradation is blocked at the same point in relation to the folded DHFR domain. **(D)** Time-course analysis of (Lum linker)-iRC degradation in *dfm1Δ* cells with an empty vector (−) or the indicated *DFM1* variants. Samples were treated and analyzed as in Fig. 2 A. iRC and FLAG-tagged Dfm1 derivatives were detected with α-HA and α-Dfm1 antibodies, respectively. Dpm1 was used as a loading control and detected with α-Dpm1 antibody. **(E)** Cysteine accessibility assay of (Lum linker)-iRC. Native lysates were obtained from WT and *dfm1Δ* cells treated with or without IAA for 90 min. Lysates were incubated with mal-PEG or with mal-PEG+1% Triton X-100, followed by α-HA coimmunoprecipitation and Western blotting using α-HA antibody. Source data are available for this figure: SourceData FS5.

Provided online are Table S1, Table S2, Table S3, and Table S4. Table S1 lists yeast strains used in this study. Table S2 lists plasmids used in this study. Table S3 lists oligonucleotides used in this study. Table S4 lists antibodies used in this study.

