## [Peer Review File · The Journal of Cell Biology]

The derlin Dfm1 couples retrotranslocation of a folded protein domain to its proteasomal degradation

Daniela Vitali, Daniel Fonseca, and Pedro Carvalho

Corresponding Author(s): Pedro Carvalho, University of Oxford

Review Timeline:

Submission Date:	2023-08-14
Editorial Decision:	2023-09-20
Revision Received:	2023-12-05
Editorial Decision:	2024-01-27
Revision Received:	2024-02-04

Monitoring Editor: Billy Tsai

Scientific Editor: Andrea Marat

Transaction Report:

DOI: <https://doi.org/10.1083/jcb.202308074>

September 20, 2023

Re: JCB manuscript #202308074

Dr. Pedro Carvalho
University of Oxford
Sir William Dunn School of Pathology
South Parks road
South Parks Road
Oxford, UK OX1 3RE
United Kingdom

Dear Dr. Carvalho,

Thank you for submitting your manuscript entitled "The derlin Dfm1 couples retrotranslocation of a folded protein domain to its proteasomal degradation". The manuscript was assessed by expert reviewers, whose comments are appended to this letter. We invite you to submit a revision if you can address the reviewers' key concerns, as outlined here.

You will see that the reviewers are somewhat mixed in their assessment. Editorially, we find that as your study has new implications for our understanding of ERAD it represents a potentially important advance and that an appropriately revised study is suitable for JCB. In revising, please respond completely to all points from reviewers 2 and 3. Regarding the concerns of reviewer 1 it is important to experimentally address the following:

1. concerns regarding the use of the Ost1 TM segment - we agree that you should use another model substrate to verify key findings
2. verify that lysosomes are not involved - this seems straightforward and should be addressed (for example via the use of lysosomal inhibitors)
7. suggests effects observed could be due to the proteasome - this should again be addressed since it seems straightforward to rule out a potential role of the proteasome

Please respond to the following points with explanations and new experimental data if necessary to respond to the reviewer's concerns:

4. questions the MWs reported and topology of insertion - if there is discrepancy regarding MW/topology, please provide an explanation
5. inconsistencies with the protease protection assays - please address this inconsistency

While we would welcome data for the following points, please use your judgment as to what is essential to experimentally address to support your main conclusions:

3. suggests there could be an intermediate with the 3xHA and opsin tags on the cytoplasmic side
6. questions the use of endoH
9. would like to know if Dfm1 and Cdc48 interact with cleavage products or full length protein
10. disagrees with the mutant Cdc48
11. suggests a further analysis of the accumulation of L fragment (and questions the gel loading)

Finally, the following points should be addressed with edits and explanations in your text:

8. disagrees with the authors interpretation of binding being negligible with questions their model - please tone down the interpretation to allow for other possibilities
12. finds the nomenclature confusing - ensure your nomenclature is consistent with published literature

While you are revising your manuscript, please also attend to the following editorial points to help expedite the publication of

your manuscript. Please direct any editorial questions to the journal office.

GENERAL GUIDELINES:

Text limits: Character count for an Article is < 40,000, not including spaces. Count includes title page, abstract, introduction, results, discussion, and acknowledgments. Count does not include materials and methods, figure legends, references, tables, or supplemental legends.

Figures: Articles may have up to 10 main text figures. Figures must be prepared according to the policies outlined in our Instructions to Authors, under Data Presentation, <https://jcb.rupress.org/site/misc/ifora.xhtml>. All figures in accepted manuscripts will be screened prior to publication.

IMPORTANT: It is JCB policy that if requested, original data images must be made available. Failure to provide original images upon request will result in unavoidable delays in publication. Please ensure that you have access to all original microscopy and blot data images before submitting your revision.

Supplemental information: There are strict limits on the allowable amount of supplemental data. Articles may have up to 5 supplemental figures. Up to 10 supplemental videos or flash animations are allowed. A summary of all supplemental material should appear at the end of the Materials and methods section.

Please note that JCB now requires authors to submit Source Data used to generate figures containing gels and Western blots with all revised manuscripts. This Source Data consists of fully uncropped and unprocessed images for each gel/blot displayed in the main and supplemental figures. Since your paper includes cropped gel and/or blot images, please be sure to provide one Source Data file for each figure that contains gels and/or blots along with your revised manuscript files. File names for Source Data figures should be alphanumeric without any spaces or special characters (i.e., SourceDataF#, where F# refers to the associated main figure number or SourceDataFS# for those associated with Supplementary figures). The lanes of the gels/blots should be labeled as they are in the associated figure, the place where cropping was applied should be marked (with a box), and molecular weight/size standards should be labeled wherever possible.

The typical timeframe for revisions is three to four months. While most universities and institutes have reopened labs and allowed researchers to begin working at nearly pre-pandemic levels, we at JCB realize that the lingering effects of the COVID-19 pandemic may still be impacting some aspects of your work, including the acquisition of equipment and reagents. Therefore, if you anticipate any difficulties in meeting this aforementioned revision time limit, please contact us and we can work with you to find an appropriate time frame for resubmission. Please note that papers are generally considered through only one revision cycle, so any revised manuscript will likely be either accepted or rejected.

Thank you for this interesting contribution to Journal of Cell Biology. You can contact us at the journal office with any questions, cellbio@rockefeller.edu or call (212) 327-8588.

Sincerely,

Billy Tsai, PhD
Monitoring Editor

Andrea L. Marat, PhD
Senior Scientific Editor

Journal of Cell Biology

Reviewer #1 (Comments to the Authors (Required)):

The manuscript by Vitali and Fonseca focuses on Dfm1, a protein proposed to be involved in the degradation of a variety of ERAD substrates. The ERAD pathway is responsible for identifying misfolded proteins in the endoplasmic reticulum (ER) and targeting them for degradation. This ensures that only properly folded and functional proteins make it to their final cellular destinations. Previous studies have implicated Dfm1 as a protein with several different functions from chaperoning integral membrane ERAD substrates to helping to dislocate the substrates into the cytoplasm. This work proposes a new function for Dfm1 in the degradation of an integral membrane protein substrate with a well-folded luminal domain.

The central idea that Dfm1 is involved in coupling retrotranslocation of transmembrane proteins that contain folded luminal domains to degradation, is interesting and does not necessarily conflict with the previously proposed functions of Dfm1. However, I think that this study falls short of demonstrating this role and does not provide significant insight into this process. I would summarize this manuscript as without Dfm1, this (possibly unique) substrate generates an interesting intermediate degradation product. How this happens is unclear.

Overall, the manuscript is clearly written. Throughout, the results are logically presented. Starting around Figure 4, the logic becomes harder to follow with experimental data contradicting aspects of the model. Overall, it is unclear to this reviewer how the results inform the authors' model.

In my opinion, there are serious flaws with the manuscript.

1. I am concerned by the use of a single substrate as the basis for all of these observations. In addition, the use of the Ost1 transmembrane segment concerns me because it might not act as a typical inert transmembrane segment. This could bring the iRC model substrate into interaction, or competition, with the OST complex itself. The cryoEM structures of the yeast and human OST complexes would indicate that this TM segment is less than ideal.
2. Is it possible that some of these fragments could be accumulating in the lysosome through ER-phagy pathways? The authors should try inhibitors of the lysosomal proteases, or deletions of key proteases, to show that the lysosome is not involved.
3. How do the authors exclude that in many of these experiments there could be an intermediate where the 3xHA and opsin tags could be on the cytoplasmic side (along with the AID sites), whereas the DHFR could be on the luminal side? This seems like it could be an explanation for some of the observations.
4. Why do all full-length proteins in Figure 5D appear to be the same molecular weight, even though the proteins differ by 64 amino acids? But in the same figure, in *dfm1D*, all proteins appear to migrate faster. Are these proteins inserted into the ER membrane with the correct topology?
5. The protease protection assays seems to have inconsistencies. The controls within these assays are not behaving as expected in key experiments (Figure 6B) and therefore, are not reliable results. Specifically, a large portion of Kar2 is accessible to proteinase K, whereas this should not be the case for a luminal protein (Figure S6A).
6. Adding to the confusion, why was endoH used instead of PNGase in 5B/6A? It is possible the S fragment is still glycosylated, but moved into the Golgi and to the lysosome, or possibly back to the ER, to generate endoH-resistant glycans? PNGase should be used.
7. Based on the model, an important part of this manuscript seems to be about the ordering of events, rather than the function of Dfm1 itself. Although, it seems possible that all of the phenotypes associated with the various mutants (*rad23Ddsk2D*, *ufd2D*) are indirect effects of inhibiting a specific pathway to the proteasome (backing up the system). Do the authors have evidence to suggest otherwise? Upon expression of iRC, what is happening with known ERAD substrates where Dfm1 is involved? And ERAD substrates without Dfm1 involved?
8. Dfm1 interacts with substrate regardless of Cdc48, and vice versa (Figure 4). They claim this is "negligible", but I would disagree. The interaction between Dfm1 and iRC is consistent in the presence of Cdc48.Del.HbYX, regardless of the state of iRC ubiquitination. This would appear to contradict my understanding of the authors' model.
9. In attempting to understand this process, it would be useful to know whether Dfm1 and Cdc48 interact with the cleavage products, or the full-length protein, both of which appear in all IP's (Figure 4A).
10. The use of the Cdc48.Del.HbYX leaves a lot to be desired. I do not agree that this mutant is likely to be what they claim it to be (Figure S5A). By their quantification, this mutant does not do what they claim and appears to have altered interactions with most cofactors. Additionally, the disruption of Ufd2 interaction is only by 50% for a highly abundant protein.
11. In Figure 3A, it is quite interesting that the cells are accumulating more of the L fragment at 25C. Even at time = 0, there is considerable accumulation for the WT cells. I understand this is likely to be because of the Ost1r1, but this is particularly important, considering lanes 1-4 were underloaded (based on the loading control), so the accumulation of the fragment may be as dramatic as the *cdc48-6* mutant. This fact was just ignored, but seems like this might be a place to gain some additional insight.
12. The use of the L/S terms for the degradation products of different sizes and origins is hard to follow. Clarifying the differences between these fragments might help readers better follow the manuscript's logic. The nomenclature is quite confusing, especially with the same naming schemes in Figure 5/6 when the fragments must be completely different. S is not equal to S, and this applies to all of the figures that have different sizes with L fragments too.

Reviewer #2 (Comments to the Authors (Required)):

The paper by Giulia Vitali et al unravels a role for Dfm1 in the degradation of folded domains following their retrotranslocation to the cytoplasm. The authors applied an auxin system to direct a model protein for ERAD in a manner that overrides the classical ERAD machinery. Using convincing biochemical analyses they identify that Dfm1 couples the retrotranslocation to the VCP module that allows the processing of folded or semi-folded luminal domains.

While I accept the interpretations of the authors, I would like to see two more important controls:

1. DHFR is used as a model for tightly folded domains, only in the presence of the folate analogs (Methotrexate or others). In the absence of these analogs DHFR is susceptible to proteinase K digestion. The authors do not use a folate analog to tightly fold the DHFR domain. Why? Without it I don't see the advantage of DHFR over GFP or other models of folded domains. I would like to see the degradation in wt cells with and without folate analogs that penetrate the ER.
2. In the absence of Dfm1, the retrotranslocated domain retains the N-terminal tag? Why the degradation is C to N? Is it due to the auxin model? I think this may raise doubts for an auxin-dependent artifact. In this respect I would urge the authors to use an ERAD substrate that is directed for degradation by the ERQC pathway and not in the auxin-mediated artificial pathway. For example it is known that CD4 when expressed in yeast is regarded as an ERAD substrate. A DHFR-CD4 fusion protein should be used to ensure that the phenotypes seen are not due to the unusual auxin-mediated pathway. I think this is a critical control.

As a minor point, the authors should discuss how do they envision the role of Dfm1 in unfolding. Is it only related to VCP or additional proteins are required? I have hard time to connect the dots.

Reviewer #3 (Comments to the Authors (Required)):

The manuscript by Vitali et al. reports a well-designed model for ERAD in cells devoid of the three known ERAD E3 ligases. Turnover to this model substrate, which consists of a stably folded DHFR domain (referred to as the retroclogger), is reminiscent of ERAD and involves several known ERAD factors. However, all known ERAD E3 ligases that are commonly assumed to form the retrotranslocon are not required. While the study unambiguously shows that the current model of a strict requirement for a protein-mediated retrotranslocon is not valid any more, the analysis of observed degradation intermediates leave room for discussion. Given the complexity of living yeast cells, likely an in depth analysis would require an in vitro system, which is clearly beyond the scope of this manuscript. Hence, the authors should take care, to accurately report the effects and indicate any limitation of their study and flag the speculative part of their model. Likewise, the authors should consider to weaken the statement in the discussion that their study "delineated the steps leading to the degradation of a membrane protein with folded luminal domain". Analysing only one artificial model substrates hardly goes along with this generalisation. Moreover, does the data not exclude that when triggered by a canonical ERAD E3 ligase, other degradation routes also exist and the auxin-induced turnover unleashed an unusual variant or redundant mechanism.

Specific points:

- although MS data of purified cleavage intermediates gives first hints about the cleavage site, indicating that the L fragment is 18 amino acids longer than the S fragment, the analysis of tryptic peptides does not allow a definite conclusion. Without the identification of the peptide terminating at the neo-C-terminus (generated by the putative proteasomal cleavage of the retrotranslocation intermediate), this analysis is immature. If the authors aim for definite answers, one could either consider a chemical enrichment protocol for C-terminal peptide or use instead reference peptides to map the approximately cleavage site.
- western blot shown in Fig. 2D does not support the notion that deletion of the Dfm1 SHP boxes "led to a strong increase in the accumulation of L fragment". Although most chase experiments show clear cut effect and therefore can be judged without quantification, this and some other more subtle effect would benefit from showing gel quantifications.
- reference to previous work on how rhomboid family proteins interact with lipids need refinement: Wang et al. and Wu et al do not make statement about lipid distortion; Ben-Shem et al. 2007 report lipid bound to the active site of the E. coli rhomboid proteases in a crystal structure that likely originates from " the mildness of the purification procedure"; instead PMID: 19278654 and 36149963 could be cited.

We would like to thank the reviewers for the constructive comments and suggestions that improved our manuscript. Below we address the various comments point-by-point.

Reviewer #1 (Comments to the Authors (Required)):

The manuscript by Vitali and Fonseca focuses on Dfm1, a protein proposed to be involved in the degradation of a variety of ERAD substrates. The ERAD pathway is responsible for identifying misfolded proteins in the endoplasmic reticulum (ER) and targeting them for degradation. This ensures that only properly folded and functional proteins make it to their final cellular destinations. Previous studies have implicated Dfm1 as a protein with several different functions from chaperoning integral membrane ERAD substrates to helping to dislocate the substrates into the cytoplasm. This work proposes a new function for Dfm1 in the degradation of an integral membrane protein substrate with a well-folded luminal domain.

The central idea that Dfm1 is involved in coupling retrotranslocation of transmembrane proteins that contain folded luminal domains to degradation, is interesting and does not necessarily conflict with the previously proposed functions of Dfm1. However, I think that this study falls short of demonstrating this role and does not provide significant insight into this process. I would summarize this manuscript as without Dfm1, this (possibly unique) substrate generates an interesting intermediate degradation product. How this happens is unclear.

Overall, the manuscript is clearly written. Throughout, the results are logically presented. Starting around Figure 4, the logic becomes harder to follow with experimental data contradicting aspects of the model. Overall, it is unclear to this reviewer how the results inform the authors' model.

In my opinion, there are serious flaws with the manuscript.

1. I am concerned by the use of a single substrate as the basis for all of these observations. In addition, the use of the Ost1 transmembrane segment concerns me because it might not act as a typical inert transmembrane segment. This could bring the iRC model substrate into interaction, or competition, with the OST complex itself. The cryoEM structures of the yeast and human OST complexes would indicate that this TM segment is less than ideal.

The presence of Ost1 transmembrane segment (TM) in iRC does not seem to limit the activity of the OST complex since the expression of iRC does not interfere with cell growth nor with glycosylation, as shown by the complete glycosylation of Pdi1, an abundant glycoprotein (Fig. 5B, 6A). Moreover, the structures of the yeast OST complex (Bai et al., Nature, 2018; Wild et al., Science, 2018) suggest that the major interactions between Ost1 and the other components of the complex are mediated by the luminal domains rather than the transmembrane segments.

Nevertheless, we followed the reviewer's suggestion considering that he/she thought that testing multiple substrates would be important to expand our conclusions to a more general mechanism. Therefore, we generated iRC(Mid2), a substrate in which the Ost1 TM has been replaced by the one of Mid2 (residues 215-260). Like Ost1, Mid2 contains a single TM that adopts a type I topology. Importantly, Mid2 does not have any stable partners in the ER given

that full length Mid2 localizes to the plasma membrane. This version of the iRC was generated as a GFP fusion protein so that its localization could be assessed by fluorescence microscopy. Like GFP-iRC, GFP-iRC(Mid2) showed a reticular distribution consistent with ER localization. In addition, auxin-induced degradation of GFP-iRC(Mid2) resulted in the appearance of L and S fragments in WT and *dfm1Δ* cells, respectively. This behavior is similar to the one observed both for iRC and GFP-iRC indicating that it is independent of the amino acid sequence in the TM region. This data is presented in a figure for the reviewer, panels A and B.

2. Is it possible that some of these fragments could be accumulating in the lysosome through ER-phagy pathways? The authors should try inhibitors of the lysosomal proteases, or deletions of key proteases, to show that the lysosome is not involved.

As suggested by the reviewer, we tested the involvement of the lysosome in iRC degradation. To this aim we analyzed cells lacking *Pep4*, a protein required for the activation of lysosomal (vacuolar) proteinases. Deletion of *pep4* in various genetic backgrounds did not affect the turnover rate of iRC or changed the pattern of fragments generated during its degradation. Therefore, together with all additional data presented in the manuscript, we are confident to conclude that iRC degradation is mediated by the proteasome and independent of lysosomes. These data are now mentioned in the main text and presented in Fig S2B.

3. How do the authors exclude that in many of these experiments there could be an intermediate where the 3xHA and opsin tags could be on the cytoplasmic side (along with the AID sites), whereas the DHFR could be on the luminal side? This seems like it could be an explanation for some of the observations.

Our data clearly showed that, in the S fragment, HA and opsin tags are exposed to the cytosol. While the HA tag is 24 residues away, the opsin tag is immediately upstream of DHFR, with only 2 amino acid residues separating it from a critical beta strand at the core of DHFR tightly folded domain. Therefore, with the opsin tag in the cytosol, there is no room for the DHFR to be in the ER lumen. Despite these constraints, we could not experimentally determine the precise topology of the S fragment in the *dfm1* mutant cells. Therefore, we developed (Lum linker)-iRC with an extended unstructured 64 amino acid linker (including 3xV5 epitope) between the tightly folded DHFR domain and the transmembrane segment. As shown in the manuscript, this substrate behaves like the original iRC but leads to an S fragment lacking the TM segment, which simplified its biochemical characterization both by protease accessibility and fractionation experiments (Figures 6B and C, respectively). To further confirm the topology of the S fragment we have now performed a cysteine-accessibility assay. Given that the L and S fragments have 2 Cys residues within the DHFR module, we used membrane-impermeable maleimide polyethylene-glycol (*mal*-PEG), a compound that reacts and covalently attaches to Cys on target proteins. Given that *mal*-PEG molecule has a molecular weight of ~5 kDa, the modified protein displays a slower migration on a Western blot. In this approach, we isolated microsomes from WT and *dfm1Δ* cells expressing (lum linker)-iRC, treated with or without auxin and incubated with *mal*-PEG, followed by anti-HA immune-precipitation to enrich the fragments. In the absence of Auxin, DHFR was not modified consistent with its localization to the ER lumen, both in WT and *dfm1Δ* cells. Upon auxin treatment, we observe that S fragment generated in *dfm1Δ* cells is robustly modified with *mal*-PEG even in the absence of triton-X100 treatment indicating that it is exposed to the cytosol. In contrast, the L fragment detected in WT cells is largely protected from *mal*-PEG modification. This result is consistent with the previous analysis by

biochemical fractionation and protease accessibility experiments and supports the notion that S fragment has been retrotranslocated and localizes to the cytosol. This additional data is now included in Figure S6E.

4. Why do all full-length proteins in Figure 5D appear to be the same molecular weight, even though the proteins differ by 64 amino acids? But in the same figure, in *dfm1Δ*, all proteins appear to migrate faster. Are these proteins inserted into the ER membrane with the correct topology?

*In figure 5D we used a 4-20% acrylamide gradient gel. The resolution of this gel in the ~100kDa region, where the various iRC proteins migrate, is insufficient to resolve small differences in molecular weight conferred by the 64 amino acids of the linker. In figure S6C, the same samples were run in a Tris/Tricine, 10% acrylamide gel which has much greater resolution. In this blot the MW differences resulting from the 64 amino acid difference between full length iRC and Lum Linker(iRC) is clearly seen. Moreover, this high resolution gel shows that besides the difference in the mw of the full length proteins, the molecular weight of the S fragments generated is the same consistent with the notion that the degradation of both substrates was blocked at the same point. This is an important result, and we modified the main text of the manuscript to make it clearer. The EndoH treatments presented in Figure 6A show that iRC and (Lum linker)-iRC are inserted with similar topology, both in WT and *dfm1Δ* cells.*

5. The protease protection assays seems to have inconsistencies. The controls within these assays are not behaving as expected in key experiments (Figure 6B) and therefore, are not reliable results. Specifically, a large portion of Kar2 is accessible to proteinase K, whereas this should not be the case for a luminal protein (Figure S6A).

*The experiment in Figures 6B is aimed at showing that the L and S fragments display distinct topology and as such behave differently upon proteinase K treatment. This is nicely demonstrated in *dfm1Δ cdc48-3* double mutant cells that accumulate both S and L fragments, generated before and after *Cdc48-3* inactivation, respectively. Remarkably, treatment of these extracts with proteinase K leads to complete digestion of the S fragment while the L fragment is protected. Given that the L and S fragments are present in the same lysate, this experiment is internally controlled. Importantly, it shows that the two fragments display distinct behavior indicative of distinct topologies. In relation to a potential discrepancy, I would like to point out that the experiments in Figures 6B and S6B were performed under different conditions. In particular, Fig 6B was performed in cells shifted to 37C in order to inactivate the temperature-sensitive *Cdc48-6* protein. We noticed that, in cells subjected to the 37C temperature shift, there was a consistent decrease in the levels of Kar2 upon incubation of membranes with proteinase K. We do not know the source of this reduction, but it can be due to a variety of reasons (for example from changes in bilayer composition due to the growth at higher temperature).*

6. Adding to the confusion, why was endoH used instead of PNGase in 5B/6A? It is possible the S fragment is still glycosylated, but moved into the Golgi and to the lysosome, or possibly back to the ER, to generate endoH-resistant glycans? PNGase should be used.

We have assessed the glycosylation of iRC and its derivatives using both PNGase and endo H. The two enzymes have produced similar results both on the full length iRC and derivatives

as well as on the fragments generated during degradation. To avoid confusion, we have added a new figure with the PNGase treatment of iRC in WT and *dfm1Δ* cells (Figure S6A).

7. Based on the model, an important part of this manuscript seems to be about the ordering of events, rather than the function of Dfm1 itself. Although, it seems possible that all of the phenotypes associated with the various mutants (rad23Ddsk2D, ufd2D) are indirect effects of inhibiting a specific pathway to the proteasome (backing up the system). Do the authors have evidence to suggest otherwise? Upon expression of iRC, what is happening with known ERAD substrates where Dfm1 is involved? And ERAD substrates without Dfm1 involved?

We are not aware of any endogenous substrates of Dfm1. However, expression of iRC does not lead to a growth defect. A variety of experiments of our lab outside of the scope of this manuscript show that iRC expression is rather innocuous and does not lead to activation of ER stress responses. In addition, the expression of iRC does not interfere with the ERAD of Erg1, an endogenous substrate of the ubiquitin ligase Doa10 (see Figs. 1G, 3D, S2A).

8. Dfm1 interacts with substrate regardless of Cdc48, and vice versa (Figure 4). They claim this is "negligible", but I would disagree. The interaction between Dfm1 and iRC is consistent in the presence of Cdc48^{AHbYX}, regardless of the state of iRC ubiquitination. This would appear to contradict my understanding of the authors' model.

Fig. 4A shows that at basal conditions (without auxin treatment), a small amount of Dfm1 and Cdc48 interact with iRC in WT cells. Upon auxin treatment, we observe an increase in iRC association with both proteins. Importantly, in Cdc48^{AHbYX} cells, the levels of Dfm1 co-immunoprecipitated with iRC is comparable to WT under basal conditions and does not change upon addition of auxin. In addition, in Fig 4B we show that mutations in Ufd2 interfere with Cdc48 association with iRC upon auxin addition. From these observations we concluded that an intact Cdc48/Ufd2 complex is required for the recruitment of Cdc48 and Dfm1 to the substrate. We have modified the text to clarify this point.

9. In attempting to understand this process, it would be useful to know whether Dfm1 and Cdc48 interact with the cleavage products, or the full-length protein, both of which appear in all IP's (Figure 4A).

The IPs were performed both in the absence of auxin, when only full length iRC is present, and ~20 min after auxin addition, when both full length and the fragments are present. Addition of auxin, which leads to iRC ubiquitination, triggers interaction of Dfm1 and Cdc48. This lead us to conclude that Cdc48 and Dfm1 recruitment is dependent on iRC ubiquitination. This in line with the current knowledge on Cdc48 function and how it is recruited to other ERAD substrates. The adaptor protein Ubx2 was shown to facilitate Cdc48 recruitment to some substrates, however we have no evidence that Ubx2 is required for iRC degradation (see Figure S4C).

At the moment we have no direct experimental evidence that Cdc48 and Dfm1 also interact with iRC derived fragments. However, our data argue that both factors remain associated with the fragments. The observation that the L fragment accumulates in cells with compromised Cdc48 ATPase function (for example Cdc48-6 and Npl4-1) indicates that Cdc48/Npl4 is essential for the clearance of this fragment, which likely requires an interaction between Cdc48 and the fragment. Similarly, Dfm1 is required for the clearance of the L fragment, likely involve binding of the two proteins.

10. The use of the Cdc48.Del.HbYX leaves a lot to be desired. I do not agree that this mutant is likely to be what they claim it to be (Figure S5A). By their quantification, this mutant does not do what they claim and appears to have altered interactions with most cofactors. Additionally, the disruption of Ufd2 interaction is only by 50% for a highly abundant protein.

It has been reported that Ufd2 interacts with the C-terminal region of Cdc48 and that it competes with Ufd3 for Cdc48 binding (Rumpf et al., 2006). Moreover, in mammalian cells the Cdc48 homologue p97 uses its very C-terminal residues to interact with the cofactors PNGase and Ufd3 (via their PUB domain). There is both biochemical and structural evidence confirming the interaction between the highly conserved Cdc48/p97 C-terminal residues and the PUB domain on various of its co-factors. Since Cdc48 has multiple functions, many of which are essential for cell viability, we attempted to be relatively conservative when generating the Cdc48^{ΔHbYX} mutant and decided to delete only the last 3 C-terminal residues. This may explain why the interaction with Ufd2 is reduced but not completely abrogated. In fact, a highly conserved acidic 3 amino acid stretch at the C-terminus of Cdc48 was left intact (see figure to reviewer 1 panel C). However, upon auxin treatment, Cdc48^{ΔHbYX} shows a clear defect in recruitment to the iRC and in promoting its degradation.

11. In Figure 3A, it is quite interesting that the cells are accumulating more of the L fragment at 25C. Even at time = 0, there is considerable accumulation for the WT cells. I understand this is likely to be because of the OsTir1, but this is particularly important, considering lanes 1-4 were underloaded (based on the loading control), so the accumulation of the fragment may be as dramatic as the cdc48-6 mutant. This fact was just ignored, but seems like this might be a place to gain some additional insight.

As mentioned by the reviewer, we observe that cells expressing OsTir1 accumulate iRC-derived fragments even in the absence of auxin. Even considering slight variation in loading between the WT and cdc48-6 cells at 25C, it seems clear to us that cdc48-6 cells accumulate higher levels of L fragment. This is observed at both temperatures but is more prominent at 37C, upon complete inactivation of Cdc48-6 protein.

12. The use of the L/S terms for the degradation products of different sizes and origins is hard to follow. Clarifying the differences between these fragments might help readers better follow the manuscript's logic. The nomenclature is quite confusing, especially with the same naming schemes in Figure 5/6 when the fragments must be completely different. S is not equal to S, and this applies to all of the figures that have different sizes with L fragments too.

We apologize if our terminology is unclear to the reviewer. We attempted many different terminologies and, in the end, this appeared to be the clearer and most straightforward. Our data show that fragments generated from the different substrates in figures 5 and 6 are conceptually the same, hence the same nomenclature. The S fragment is the product of the degradation of everything on the C-terminal side of DHFR and it retains the transmembrane segment only when it is close enough to the DHFR module. The same logic goes for the L fragment, which is generated by the degradation of everything C-terminus of the transmembrane segment.

Reviewer #2 (Comments to the Authors (Required)):

The paper by Giulia Vitali et al unravels a role for Dfm1 in the degradation of folded domains following their retrotranslocation to the cytoplasm. The authors applied an auxin system to direct a model protein for ERAD in a manner that overrides the classical ERAD machinery. Using convincing biochemical analyses they identify that Dfm1 couples the retrotranslocation to the VCP module that allows the processing of folded or semi-folded luminal domains.

While I accept the interpretations of the authors, I would like to see two more important controls:

1. DHFR is used as a model for tightly folded domains, only in the presence of the folate analogs (Methotrexate or others). In the absence of these analogs DHFR is susceptible to proteinase K digestion. The authors do not use a folate analog to tightly fold the DHFR domain. Why? Without it I don't see the advantage of DHFR over GFP or other models of folded domains. I would like to see the degradation in wt cells with and without folate analogs that penetrate the ER.

It is well established that folding of mammalian DHFR is stabilized by the presence of folate analogues such as methotrexate. However, in our study we use E. coli DHFR, which folds more rapidly and has a more stable folding irrespective of the presence of folate ligands (Lee et al., 2001). Previous studies on ERAD took advantage of these properties of E. coli DHFR and also did show that the use of folate analogues was dispensable (Bhamidipati et al., 2005). We have clarified this point in the text to avoid any confusion.

2. In the absence of Dfm1, the retrotranslocated domain retains the N-terminal tag? Why the degradation is C to N? Is it due to the auxin model? I think this may raise doubts for an auxin-dependent artifact. In this respect I would urge the authors to use an ERAD substrate that is directed for degradation by the ERQC pathway and not in the auxin-mediated artificial pathway. For example it is known that CD4 when expressed in yeast is regarded as an ERAD substrate. A DHFR-CD4 fusion protein should be used to ensure that the phenotypes seen are not due to the unusual auxin-mediated pathway. I think this is a critical control.

We have been using the auxin-inducible degradation system to uncouple targeting and folding of the substrate from its degradation. This allows us to specifically focus on membrane proteins with folded luminal domain without concerns about fractions of the substrate not properly inserted in the membrane or not correctly folded. Therefore, testing a different substrate lacking an inducible degradation system would deviate from the focus of this work. Although we cannot exclude that the C to N degradation of iRC might be due to the AID system and that we should be careful in generalizing, the system we have been using would resemble physiological condition where an endogenous substrate with the same topology would be ubiquitinated on the cytosolic domain triggering its degradation. We envision that also in such situation the degradation would be from C to N terminus.

As a minor point, the authors should discuss how do they envision the role of Dfm1 in unfolding. Is it only related to VCP or additional proteins are required? I have hard time to connect the dots.

According to our data, we envision that Cdc48 generates the pulling force and allows the substrate to engage with Dfm1, which would mediate the unfolding. However, we cannot

exclude the requirement of other proteins cooperating with Dfm1 in unfolding the luminal domain and future studies will be required to clarify this point. We have changed the discussion to clarify these point.

Reviewer #3 (Comments to the Authors (Required)):

The manuscript by Vitali et al. reports a well-designed model for ERAD in cells devoid of the three known ERAD E3 ligases. Turnover to this model substrate, which consists of a stably folded DHFR domain (referred to as the retroclogger), is reminiscent of ERAD and involves several known ERAD factors. However, all known ERAD E3 ligases that are commonly assumed to form the retrotranslocon are not required. While the study unambiguously shows that the current model of a strict requirement for a protein-mediated retrotranslocon is not valid any more, the analysis of observed degradation intermediates leave room for discussion. Given the complexity of living yeast cells, likely an in depth analysis would require an in vitro system, which is clearly beyond the scope of this manuscript. Hence, the authors should take care, to accurately report the effects and indicate any limitation of their study and flag the speculative part of their model. Likewise, the authors should consider to weaken the statement in the discussion that their study "delineated the steps leading to the degradation of a membrane protein with folded luminal domain". Analysing only one artificial model substrates hardly goes along with this this generalisation. Moreover, does the data not exclude that when triggered by a canonical ERAD E3 ligase, other degradation routes also exist and the auxin-induced turnover unleashed an unusual variant or redundant mechanism.

We thank the reviewer for the suggestion. We did not mean to overstate the results and the specific sentence was changed to make avoid any ambiguity.

Specific points:

- although MS data of purified cleavage intermediates gives first hints about the cleavage site, indicating that the L fragment is 18 amino acids longer than the S fragment, the analysis of tryptic peptides does not allow a definite conclusion. Without the identification of the peptide terminating at the neo-C-terminus (generated by the putative proteasomal cleavage of the retrotranslocation intermediate), this analysis is immature. If the authors aim for definite answers, one could either consider a chemical enrichment protocol for C-terminal peptide or use instead reference peptides to map the approximately cleavage site.

Based on our data we don't think that there is a specific cleavage site. In fact, the L fragment often appears as a smear on the SDS-Page and we think that this is due to the presence of different species of fragment cleaved at slightly different sites. On the other end, the S fragment seems to be more defined, probably because the cleavage always happens at a specific distance from DHFR. By MS we could detect an 18 amino acids difference between the L and S fragment, but we do not think that represent the exact cleavage site.

- western blot shown in Fig. 2D does not support the notion that deletion of the Dfm1 SHP boxes "led to a strong increase in the accumulation of L fragment". Although most chase experiments show clear cut effect and therefore can be judged without

quantification, this and some other more subtle effect would benefit from showing gel quantifications.

We have included the quantification of the L fragment in Fig. 2F.

- reference to previous work on how rhomboid family proteins interact with lipids need refinement: Wang et al. and Wu et al do not make statement about lipid distortion; Ben-Shem et al. 2007 report lipid bound to the active site of the E. coli rhomboid proteases in a crystal structure that likely originates from " the mildness of the purification procedure"; instead PMID: 19278654 and 36149963 could be cited.

We thank reviewer for the suggestion. We changed the text and added the references to more accurately cite previous studies.

(A) Fluorescent and DIC microscopy images of WT and *dfm1* Δ cells expressing GFP-tagged iRC(Mid2), an iRC version with the transmembrane segment of Mid2.

(B) WT and *dfm1* Δ cells expressing GFP-tagged versions of iRC or iRC(Mid2) were treated with IAA and CHX and collected at the indicated time points. Whole cell lysates were analysed by SDS-PAGE and western blot with the indicated antibodies as described in manuscript.

Δ

(C) Evolutionary conservation analysis of the C-terminus (821-835 aa) of Cdc48 using ConSurf (Ashkenazy et al. Nucleic Acids Research, 2016)

January 27, 2024

RE: JCB Manuscript #202308074R

Dr. Pedro Carvalho
University of Oxford
Sir William Dunn School of Pathology
South Parks road
South Parks Road
Oxford, UK OX1 3RE
United Kingdom

Dear Dr. Carvalho:

Thank you for submitting your revised manuscript entitled "The derlin Dfm1 couples retrotranslocation of a folded protein domain to its proteasomal degradation". You will see that while reviewer 3 fully supports publication, the other two reviewers have raised some remaining concerns. We appreciate reviewer 2's feedback regarding their experience with the DHFR domain. While this was a concern in their original review, as you have cited published words supporting your use of *E. coli* DHFR we accept your response based on the literature. Therefore, overall we find you have sufficiently addressed all essential concerns from the first editorial decision, and we would be happy to publish your paper in JCB pending final revisions necessary to meet our formatting guidelines (see details below). In your final revision you must also respond to the remaining concerns of reviewer 1 with text edits to account for other interpretations.

A. MANUSCRIPT ORGANIZATION AND FORMATTING:

- 1) Text limits: Character count for Articles is < 40,000, not including spaces. Count includes abstract, introduction, results, discussion, and acknowledgments. Count does not include title page, figure legends, materials and methods, references, tables, or supplemental legends.
- 2) Figures limits: Articles may have up to 10 main text figures.
- 3) Figure formatting: Scale bars must be present on all microscopy images, including inset magnifications. Molecular weight or nucleic acid size markers must be included on all gel electrophoresis.
- 4) Statistical analysis: Error bars on graphic representations of numerical data must be clearly described in the figure legend. The number of independent data points (n) represented in a graph must be indicated in the legend. Statistical methods should be explained in full in the materials and methods. For figures presenting pooled data the statistical measure should be defined in the figure legends. Please also be sure to indicate the statistical tests used in each of your experiments (either in the figure legend itself or in a separate methods section) as well as the parameters of the test (for example, if you ran a t-test, please indicate if it was one- or two-sided, etc.). Also, if you used parametric tests, please indicate if the data distribution was tested for normality (and if so, how). If not, you must state something to the effect that "Data distribution was assumed to be normal but this was not formally tested."
- 5) Abstract and title: The abstract should be no longer than 160 words and should communicate the significance of the paper for a general audience. The title should be less than 100 characters including spaces. Make the title concise but accessible to a general readership.
- 6) Materials and methods: Should be comprehensive and not simply reference a previous publication for details on how an experiment was performed. Please provide full descriptions in the text for readers who may not have access to referenced manuscripts.
- 7) **** All antibodies, cell lines, animals, and tools used in the manuscript should be described in full, including accession numbers for materials available in a public repository such as the Resource Identification Portal. Please be sure to provide the sequences for all of your primers/oligos and RNAi constructs in the materials and methods. You must also indicate in the methods the source, species, and catalog numbers (where appropriate) for all of your antibodies. Please also indicate the acquisition and quantification methods for immunoblotting/western blots. ****

8) Microscope image acquisition: The following information must be provided about the acquisition and processing of images:

- a. Make and model of microscope
- b. Type, magnification, and numerical aperture of the objective lenses
- c. Temperature
- d. Imaging medium
- e. Fluorochromes
- f. Camera make and model
- g. Acquisition software
- h. Any software used for image processing subsequent to data acquisition. Please include details and types of operations involved (e.g., type of deconvolution, 3D reconstitutions, surface or volume rendering, gamma adjustments, etc.).

10) Supplemental materials: There are strict limits on the allowable amount of supplemental data. Articles may have up to 5 supplemental figures. Please also note that tables, like figures, should be provided as individual, editable files. A summary of all supplemental material should appear at the end of the Materials and methods section.

13) ORCID IDs: ORCID IDs are unique identifiers allowing researchers to create a record of their various scholarly contributions in a single place. Please note that ORCID IDs are now *required* for all authors. At resubmission of your final files, please be sure to provide your ORCID ID and those of all co-authors.

Please note that JCB now requires authors to submit Source Data used to generate figures containing gels and Western blots with all revised manuscripts. This Source Data consists of fully uncropped and unprocessed images for each gel/blot displayed in the main and supplemental figures. Since your paper includes cropped gel and/or blot images, please be sure to provide one Source Data file for each figure that contains gels and/or blots along with your revised manuscript files. File names for Source Data figures should be alphanumeric without any spaces or special characters (i.e., SourceDataF#, where F# refers to the associated main figure number or SourceDataFS# for those associated with Supplementary figures). The lanes of the gels/blots should be labeled as they are in the associated figure, the place where cropping was applied should be marked (with a box), and molecular weight/size standards should be labeled wherever possible.

Journal of Cell Biology now requires a data availability statement for all research article submissions. These statements will be published in the article directly above the Acknowledgments. The statement should address all data underlying the research presented in the manuscript. Please visit the JCB instructions for authors for guidelines and examples of statements at (<https://rupress.org/jcb/pages/editorial-policies#data-availability-statement>).

B. FINAL FILES:

-- High-resolution figure and MP4 video files: See our detailed guidelines for preparing your production-ready images,

<https://jcb.rupress.org/fig-vid-guidelines>.

Thank you for this interesting contribution, we look forward to publishing your paper in Journal of Cell Biology.

Sincerely,

Billy Tsai, PhD
Monitoring Editor

Andrea L. Marat, PhD
Senior Scientific Editor

Journal of Cell Biology

Reviewer #1 (Comments to the Authors (Required)):

Overall, this manuscript describes an interesting phenomenon, but in my opinion, fails to provide enough insight to support the title. My specific points that the editors requested to be addressed by the authors (1,2,7) have been mostly addressed.

I disagree with the response to the original point 6. The fact that the Kar2 control is not behaving means whatever interpretations the authors have made are suspect (50% Kar2 degraded without TX). I'm not suggesting this is correct, but an alternate explanation of these results could be that there may be distinct populations of cells containing L and S fragments, and only those with the S fragment were lysed. I think the authors should at least address the experimental problem with Kar2 being accessible to PK in the text of the manuscript, rather than to ignore it. I think broadly, other interpretations should be discussed where they have not been ruled out.

With point 12, the nomenclature of the S/L, maybe consider S(Ost1) or S-Ost1 (with Ost1 as a subscript).

Reviewer #2 (Comments to the Authors (Required)):

I would like to see that the DHFR domain used here is proteinase K resistant. This is a must do control. All the interpretations rely on the existence of a tightly folded domain, which in our experience relies on the presence of a folate analogue.

Reviewer #3 (Comments to the Authors (Required)):

The changes made in the revised version improve the manuscript. I have no further comments and fully support publication of this interesting study.